



# LONG-TERM MEASUREMENT OF SUB-3NM PARTICLES AND THEIR PRECURSOR GASES IN THE BOREAL FOREST

Juha Sulo[1], Nina Sarnela[1], Jenni Kontkanen[1], Lauri Ahonen[1], Pauli Paasonen[1], Tiia Laurila[1], Tuija Jokinen[1], Juha Kangasluoma[1,2], Tuukka Petäjä[1], Markku Kulmala[1,2] and Katrianne Lehtipalo[1,3]

[1]Institute for Atmospheric and Earth System Research/Physics, University of Helsinki, 00014 Helsinki, Finland
[2]Aerosol and Haze Laboratory, Beijing Advanced Innovation Center for Soft Matter Science and Engineering, Beijing University of Chemical Technology, Beijing, China
[3]Finnish Meteorological Institute, 00560 Helsinki, Finland

*Correspondence to*: Juha Sulo (juha.sulo@helsinki.fi)

**Abstract.** The knowledge of the dynamics of sub-3nm particles in the atmosphere is crucial for our understanding of first steps of atmospheric new particle formation. Therefore, accurate and stable long-term measurements of the smallest atmospheric particles are needed. In this study, we analyzed over five years of particle concentrations in size classes 1.1–1.3 nm, 1.3–1.7 nm and 1.7–2.5 nm obtained with the Particle Size Magnifier (PSM) and three years of precursor vapor concentrations measured with the Chemical Ionization Atmospheric Pressure Interface Time-of-Flight mass spectrometer (CI-APi-ToF) at the SMEAR II station in Hyytiälä, Finland. The results show that the 1.1–1.3 nm particle concentrations have a daytime maximum during all seasons, which is due to increased photochemical activity. There are significant seasonal differences in median concentrations of 1.3–1.7 nm and 1.7–2.5 nm particles, underlining the different frequency of new particle formation between seasons. In particular, concentrations of 1.3 – 1.7 nm and 1.7 – 2.5 nm particles are notably higher in spring than during other seasons. Aerosol precursor vapors have notable diurnal and seasonal differences as well. Sulfuric acid and highly oxygenated organic molecule (HOM) monomer concentrations have clear daytime maxima, while HOM dimers have their maxima during the night. HOM concentrations for both monomers and dimers are the highest during summer and the lowest during winter. Higher median concentrations during summer result from increased biogenic activity in the surrounding forest. Sulfuric acid concentrations are the highest during spring and summer, with autumn and winter concentrations being two to three times lower. A correlation analysis between the sub-3nm concentrations and aerosol precursor vapor concentrations indicates that HOMs, particularly their dimers, and sulfuric acid play a significant role in new particle formation in the boreal forest. Our analysis also suggests that there might be seasonal differences in new particle formation pathways that need to be investigated further.



## 1 Introduction

Atmospheric aerosols are one of the largest sources of uncertainty in climate models. To diminish these uncertainties, it is vital to understand the sources and formation pathways of aerosol particles. New particle formation (NPF) in the atmosphere has been observed to occur in various environments (Kulmala et al., 2004; Kerminen et al., 2018 and references therein) from megacities (Hofman et al., 2016; Wu et al., 2007; Qi et al., 2017) to rainforests (Wimmer et al., 2018; Andreae et al., 2018), to rural areas (Mäkelä et al., 1997; Vakkari et al., 2011; Heintzenberg et al., 2017; Nieminen et al., 2014) and even

polar areas (Weller et al., 2015; Kyrö et al., 2013; Järvinen et al., 2013; Sipilä et al., 2016). In the laboratory, aerosol particles have been observed to form through various pathways involving different chemical compounds and ions (e.g. Kirkby et al., 2016; Lehtipalo et al., 2018). Based on various estimates (Merikanto et al., 2009; Dunne et al., 2016; Kulmala et al., 2016), 40-80% of aerosol particles in the atmosphere are formed from condensing vapours and thus NPF contributes significantly to aerosol number concentrations in the atmosphere.


It is widely understood that sulfuric acid (SA) plays a significant role in atmospheric NPF (Kulmala et al., 2006; Riipinen et al., 2007; Paasonen et al., 2010). However, SA alone is not enough to explain atmospheric observations in the boundary layer (Chen et al., 2012; Paasonen et al., 2012) Studies indicate that at least one more vapor, for example a base like amine or ammonia, is needed for stabilizing the growing clusters (Almeida et al., 2013; Kirkby et al., 2016; Kürten et al., 2018).

Highly oxygenated organic molecules (HOMs), formed in the atmosphere through auto-oxidation from volatile organic compounds, likely partake in particle growth everywhere in the boundary layer (Ehn et al. 2014; Mohr et al., 2019). Some HOMs can form charged clusters on their own, but it is unclear how important these pure biogenic clusters are for atmospheric NPF (Kirkby et al., 2016; Rose et al., 2018; Bianchi et al., 2017).

Freshly formed particles have typical sizes of 1–2 nm in diameter and they grow to larger sizes by condensation of low volatility vapors (Kulmala et al., 2014). Measuring these sub-3nm particles is critical to a proper understanding of NPF and the early steps of particle growth in the atmosphere. One method of measuring sub-3nm particles is the Particle Size Magnifier (PSM) (Vanhanen et al., 2011). The PSM is a two-stage condensation particle counter (CPC) used to grow and count sub-3nm particles. Sub-3nm particles have been measured with the PSM in various environments (Kontkanen et al.,

2017), from polluted Chinese megacities (Xiao et al., 2015) to rural areas (Kulmala et al., 2013) and mountain tops (Rose et al., 2015). Measurement devices are prone to errors, particularly in the field, and understanding the factors that affect the performance of the PSM and validating data from PSM measurements is an ongoing challenge (Kangasluoma et al., 2013; Kangasluoma et al., 2014; Kangasluoma et al., 2016b).

In this study, we first investigate what are the optimal settings to operate the PSM at the SMEAR II station in Hyytiälä, Southern Finland, to detect sub-3nm particles. Then, we analyze the over 5-year dataset using these optimized settings and





investigate the diurnal and seasonal variation of sub-3nm particle concentrations and their connection to atmospheric NPF. Additionally, the sub-3nm particle concentrations are compared with some of their possible precursor vapor concentrations in order to determine which vapors participate in atmospheric NPF.

## 2 Materials and methods

### 2.1 Measurement location

All measurements were conducted at the SMEAR II (Station for Measuring Forest Ecosystem Atmosphere Relations) station in Hyytiälä, Southern Finland (61∘5' 0'' N, 24∘170' E; 181 m above sea level) (Hari and Kulmala, 2005) between April 2014 and April 2020. Hyytiälä is roughly 200 kilometers from Helsinki, with the closest urban center Tampere about 60 kilometers to the southwest. Tampere has a population of roughly 230 000 people. The SMEAR II station is considered a rural background station for atmospheric measurements and is surrounded by a Scots pine (Pinus sylvestris) forest. The emissions of volatile organic compounds at the station are dominated by biogenic vapors from the surrounding forest, monoterpenes in particular (Rantala et al., 2015). NPF events including particle growth have been observed only during daytime (Buenrostro Mazon et al., 2016), while biogenic cluster formation is common also during evening and night (Rose et al., 2018).

### 2.2 Particle size magnifier

We used the Particle Size Magnifier (PSM) to measure the concentrations of sub-3nm particles. The A11 nCNC –system (nano Condensation Nucleus Counter) is a particle counter system developed for measuring particle concentrations larger than 1 nm in size and size distributions in the sub – 3nm particle size region (Vanhanen et al., 2011). In this system, the PSM operates as a pre-conditioner, in which the small particles are first grown before they are funneled into a CPC for further growth and optical detection. The PSMs used in this study were manufactured by Airmodus (model A10) and used together with an Airmodus A20 CPC. In the PSM, the sample flow is turbulently mixed in the mixing region with a saturated heated flow to achieve supersaturation in the growth tube. Diethylene glycol (DEG) starts to condense on the sample particles and they are grown up to the size of circa 100 nm in the cooled growth tube. From the growth tube, the flow is directed to a CPC where the particles are grown with butanol to optically detectable sizes.

Particle activation within the PSM depends on the supersaturation level in the PSM, which in turn depends on the temperature difference between the heated saturated flow, the sample flow, and the growth tube, as well as the saturator flow rate (Kangasluoma et al,. 2016b; Vanhanen et al., 2011; Okuyama et al., 1984). Because of that, the supersaturation can be adjusted by changing the saturator flow rate or the temperature difference between these components. A larger temperature difference between the heated saturated flow and the growth tube or a higher saturator flow rate leads to a higher





supersaturation level, which means that smaller particles are activated and thus the cutoff size of the instrument, i.e. the diameter at which 50% of the particles are activated, is lower. However, too high a supersaturation level will lead to
formation of droplets via homogeneous nucleation causing a background signal. Studies have also shown that the cutoff size for particles depends not only on the supersaturation, but on the chemical composition of the sample particles, particle charge and the condensing fluid (see Sect 2.2.3.) (Kangasluoma et al., 2016b; Winkler et al., 2012).

The PSM can be run in several measurement modes, but in this study, we only used the scanning mode. The scanning mode
makes it possible to determine the particle size distribution of sub-3nm particles. In the scanning mode, the PSM scans through saturator flow rates to obtain the particle size distribution. In our study, the PSM scanned from 0.1 liters per minute (lpm) to 1.3 lpm and back to 0.1 lpm in four minutes. During the scan, particle counts are saved at a one second interval to the raw data file. Each measurement corresponds to the total concentration of particles above a certain size determined by the saturator flow rate, and therefore the changes in the measured concentration during a scan can be used to acquire information
of the size distribution (Lehtipalo et al. 2014).

The PSM measurements were conducted at the SMEAR II station in a container at ground level. Due to maintenance, the instrument was changed on 23 February 2017 and 1 October 2018. In 2014, the sampling was done with a 40 cm long stainless-steel tube with 2.5 lpm flow rate. The diffusion losses were corrected for in the data inversion. From 2015 onward,
the sampling was done with the core sampling method, where the air was taken from the outside at 7.5 lpm through a 40 cm stainless steel tube and the sample was taken from the middle of the flow into the PSM at 2.5 lpm to minimize sampling losses. The inlet also included a mechanism which allowed the bypass flow to be briefly reversed in order to provide particle-free air in ambient relative humidity for PSM background measurements (Kangasluoma et al., 2016b).

### 2.2.1 PSM data analysis

Before the PSM raw data was inverted, the data was run through a quality control algorithm in which we assumed that the measured total concentration during each scan should have a positive and statistically significant ($p < 0.05$) correlation with the PSM saturator flow rate (Chan et al., 2020). Increasing the flow rate should always result in increased total concentration because we are activating a larger size range of particles. We omitted scans not fulfilling this requirement from further analysis, as they would lead to a negative concentration of sub-3nm particles. These so called "bad scans" could result from
air mass changes or other variations in aerosol concentrations that are faster than the scan time, or if the concentration of sub-3nm particles is so low that it cannot be detected with this method.

We inverted the raw PSM data with the kernel method described in (Lehtipalo et. al., 2014) to obtain the size distribution of sub-3nm particles. The kernel method uses a non-negative matrix inversion routine to calculate the size distribution based on
activation curves measured during the calibration of the instrument. The data was inverted to a 16-minute resolution (4



scans) and three size bins (1.1 – 1.3 nm 1.3 – 1.7 nm, 1.7 – 2.5 nm) to minimize the effect of noise on the analysis, but still retain a high enough time resolution for the analysis.

Recently, Cai et al. (2019) recommended another inversion routine for PSM data, the expectation-maximization algorithm
(EM).  However, as our data was already inverted with the kernel method and the EM method is computationally expensive, we decided to stick with the kernel method. The two inversion methods produce similar concentrations and size distributions when both are optimized for the dataset in question (Cai et al., 2019). It remains future work to test the applicability of the EM algorithm for SMEAR II data and optimize it to conditions with rather low particle concentrations.

**2.2.2 Effect of supersaturation and background counts**

At optimal temperature settings the PSM should activate a large fraction of even the smallest particles around 1 nm, while still minimizing the effect of homogenous nucleation within the PSM. In practice, a small background from homogeneous nucleation needs to be tolerated at higher saturator flow rates in order to activate the smallest particles, especially organic
clusters. The amount of homogeneously nucleated droplets can be taken as an indicator of the supersaturation level (activation efficiency) (Jiang et al., 2011).

To monitor the instrument operation and supersaturation level, the background counts were automatically measured three times a day.. Due to changes in external conditions and the state of the instrument, the background varies and if the operator
though it was too high (> ca. 50 $cm^{-3}$) or too low (< ca. 1 $cm^{-3}$), the temperature settings were adjusted to keep the cut-off sizes same as before. During the whole measurement period, the daily averaged (over all saturator flow rates) homogenous background varied from less than 1 $cm^{-3}$ to almost a 1000 $cm^{-3}$. The background counts were subtracted from the data during data processing.

We investigated the fraction of scans discarded during data quality control as a function of the background concentrations (daily median) (Figure 1). The u-shape of the bad scan percentage clearly shows that the quality of the scans goes down if the PSM background is either too low (<0.1 – 1) or too high (>10).  If the supersaturation level and consequently the background level inside the PSM is too low, the smallest particles cannot be activated and there is no detectable signal, which leads to noisy scans. A high background, on the other hand, can indicate that the PSM is not functioning properly.
Thus, based on the quality of the scans alone, the PSM appears to work most stably when the background signal at the highest saturator flow rate is between 1 and 10 $cm^{-3}$. However, in our measurements, the devices were never intentionally run at background levels higher than circa 50 $cm^{-3}$. For this reason, the PSM could be stable at higher background levels as well, but our data does not allow us to draw conclusions on that. Furthermore, this behavior seems to be uncorrelated with





the measured relative humidity at the measurement location, although laboratory studies have shown that RH can affect the

particle activation efficiency with DEG (Kangasluoma et al., 2013; Jiang et al., 2011).

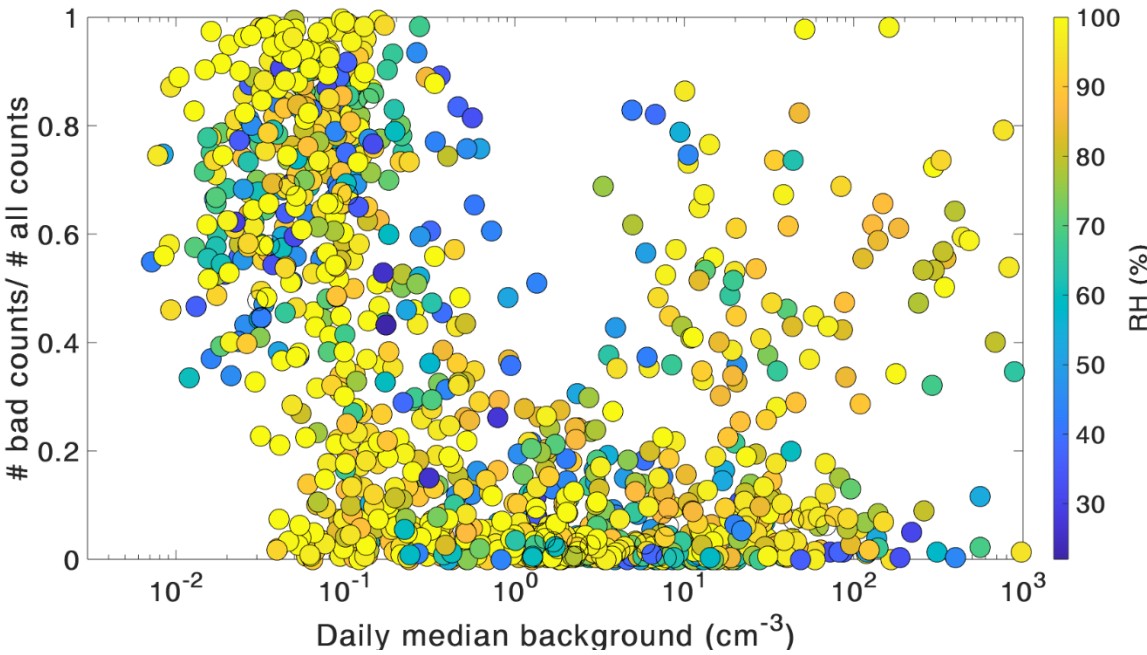

**Figure 1: The fraction of scans discarded during data quality check plotted against the daily medians of PSM background concentration. The color of the circles shows the ambient relative humidity (RH).**


To investigate the effect of the background level (supersaturation) on the activation of the smallest particles, we split the data points to groups where the background is below 1 cm$^{-3}$, above 10 cm$^{-3}$ or between these two limits. The limits were chosen based on the observed fraction of bad scans in Fig. 1. We then studied 1.1 – 1.3 nm concentration as a function of the PSM background (Figure 2). When the background is under 1 cm$^{-3}$, the measured concentrations are on average notably lower

than when the background level is above 1 cm$^{-3}$, indicating that we are not activating all of the 1.1 - 1.3 nm particles at those settings. However, if the background level rises to over 10 cm$^{-3}$, the variation becomes bigger and the median concentration smaller, underlining the various factors affecting the concentration at higher background. At very high background levels the PSM likely activates large vapor molecules or clusters whose concentrations are not stable, leading to larger variation in the concentration. When these species dominate the activated particles, the particle size distribution cannot be easily resolved

from the scans. Other causes for higher variation at high background include but are not necessarily limited to faulty instrumentation, dirty sample lines and dominant homogeneous nucleation of the working fluid.

The analysis described above lead to the conclusion that in the conditions of the SMEAR II station, the optimal settings for the PSM is found when the measured background is between 1 cm$^{-3}$ and 10 cm$^{-3}$. As mentioned before, the PSM could be





run stably at background levels above 10 cm$^{-3}$ as well, but due to the results discussed above, we selected for further analysis only PSM data with the instrument background between 1 and 10 cm$^{-3}$.

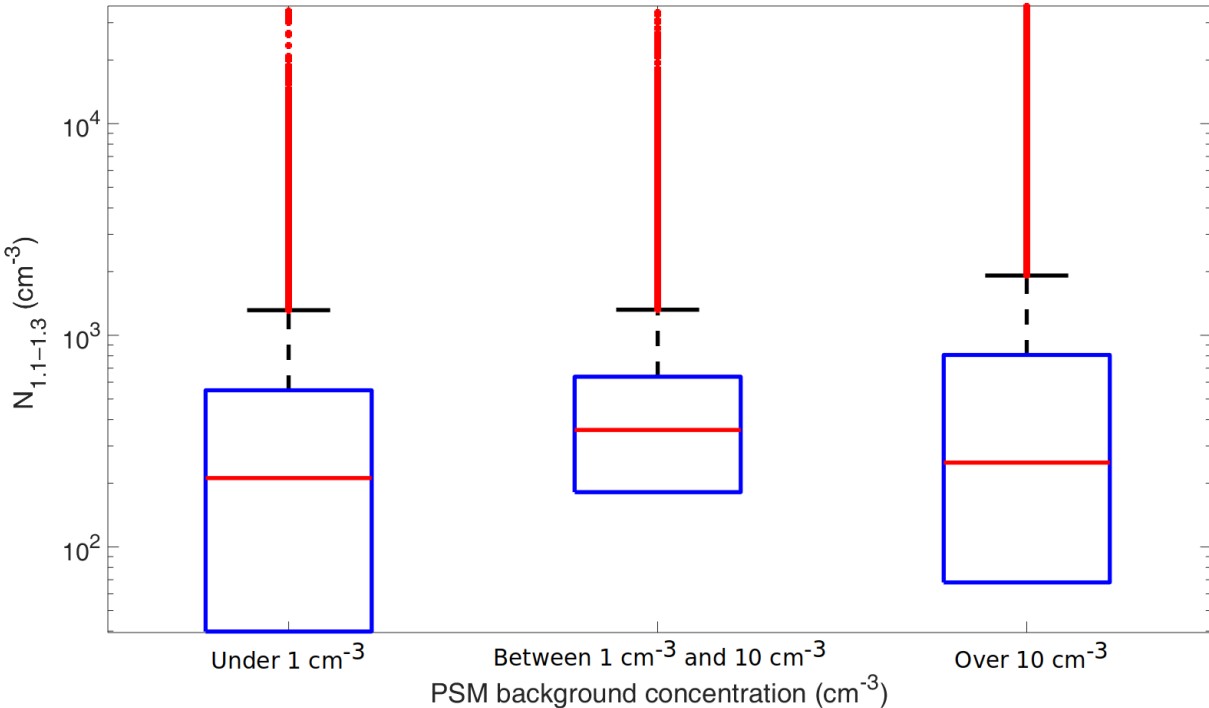

**Figure 2: Box plots of the measured 1.15 – 1.3 nm particle concentration at SMEAR II grouped with the PSM background**
**concentration. The red line is the median concentration and the blue box contains 50% (25th to 75th percentile) of all data points. The whiskers mark the location of the 95th and 5th percentile data points and the red plusses are outliers beyond those percentiles.**

### 2.2.3 Measurement uncertainties

Measuring sub-3nm particles involves notable uncertainties, as small particles are very difficult to detect. Both PSM
calibration and measurement are sensitive to the chemical composition of the particles being measured. The activation probability with DEG seems to be lower for organic particles than for inorganic particles. The cutoff size, the diameter at which 50% of the particles are activated in the PSM, can be over a nanometer larger for organic particles than for inorganic particles (Kangasluoma et al., 2014, Kangasluoma et al., 2016b), but there is no systematic studies for different kinds of ambient particles. Because we do not know the exact chemical composition of the particles in the ambient air, the sizing in
the measurement contains uncertainties. In addition, the activation probability of particles is also slightly different for charged and non-charged (neutral) particles (Kangasluoma et al., 2016b).





The PSM is calibrated by measuring particles from a known source. A certain particle size is selected with a differential mobility analyzer (DMA) and an electrometer is used as a reference instrument. This gives us the PSM detection efficiency

for each selected size. The PSMs in this study were calibrated using charged tungsten oxide particles in the size range between 1.0 and 3.2 nm in mobility diameter, as there is no good calibration method and reference instrument readily available for neutral particles. Therefore, the diameters given should be taken as activation-equivalent sizes (we assume that the particles would activate as charged tungsten oxide particles do). The PSM may also be sensitive to ambient conditions, mainly relative humidity (Kangasluoma et al, 2013; Jiang et al., 2011). More discussion on the uncertainties can be found in

Kangasluoma et al. (2020) and references therein.

When dealing with long time series, an additional complication arises from changing and maintaining equipment. While the different PSMs used in the study over the years are essentially similar devices, they have slightly varying cutoff limits and detection efficiency curves, which has been taken into account during data processing, but which could still affect the final

inverted concentrations. The data preprocessing and inversion method can also produce additional uncertainties which are difficult to quantify (Lehtipalo et al., 2014; Cai et al., 2019).

To estimate the magnitude of error caused by the uncertainties related to PSM measurement, we compared the ion concentrations detected by the PSM to those from a Neutral Cluster and Air Ion Spectrometer (NAIS, described briefly in

section 2.4), which is the only other instruments measuring in the same size range at SMEAR II. The ion concentrations were acquired from a PSM with an ion trap inlet (Wagner et al., 2017, Kangasluoma et al., 2016a). The setup is otherwise similar to the PSM used in the rest of this study, but the ion trap is switched on every 8 minutes and then off again after 8 minutes. This allows us to differentiate between neutral particle and total particle concentrations and acquire the ambient ion concentration from the PSM.


The comparison shows that the PSM detects small ions and charged particles fairly well and the overall concentrations measured with the PSM and the NAIS are of the same magnitude in the size range used in this study (Figure 3). The relative concentration (from binned y-directional medians) measured by the PSM is between 29 and 100 %, with the median relative concentration being 65 %. This suggests that most, but not all of the ambient ions are activated by the PSM, and

that we are more likely underestimating than overestimating the ambient concentrations of small particles. It should be noted that the ions are a mixture of different chemical compounds, and the ion composition varies diurnally and seasonally in boreal forest (Ehn et al. 2010), so we expect that their activation fraction (relative to tungsten oxide ions used for calibrating the PSM), varies as well. Detailed analysis of the activation of different types of ambient ions in the PSM is subject to future studies. Based on laboratory experiments which conclude that the effect of charge on the cutoff size is between 0 and 0.50

nm (Kangasluoma et al., 2016b), the above results mean that we likely activate neutral particles as well.



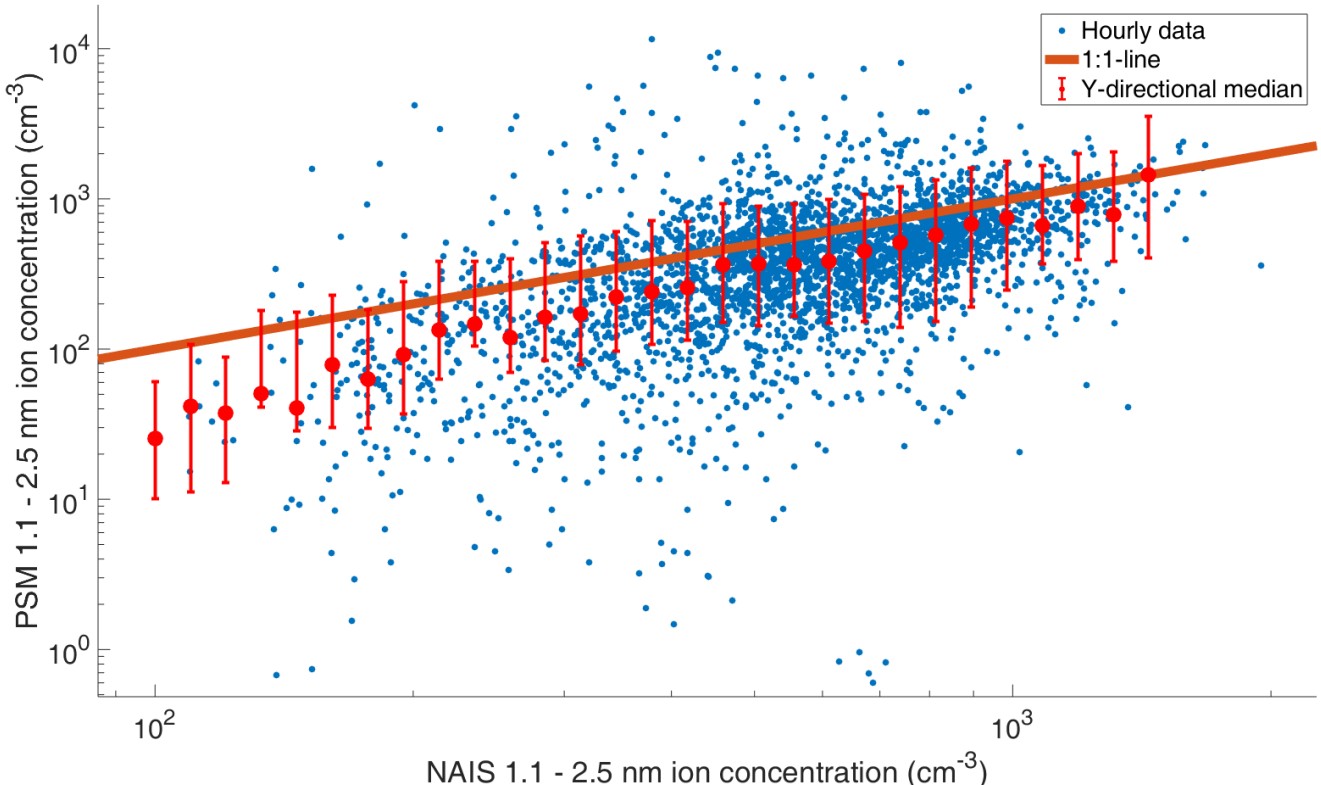

**Figure 3: Comparison of 1.1 – 2.5 nm ambient ions measured by PSM and NAIS at SMEAR II. The orange line is the 1:1 line, the blue data points are hourly medians and the red points are lognormally binned y-directional medians with the red whiskers showing the 75th and 25th percentile.**


### 2.3 CI-APi-ToF

We used Chemical Ionization Atmospheric Pressure interface Time of Flight Mass Spectrometer (CI-APi-TOF, Jokinen et al., 2012) with nitrate ion as a reagent ion to measure low-volatility vapors at the SMEAR II station. The detection of CI-

APi-TOF is based on proton transfer reactions or clustering with the nitrate ion through collisions with nitrate ions ($NO3-$), nitrate ion-nitric acid dimers ($HNO_3NO_3^-$) and nitrate ion-nitric acid trimers (($HNO_3$)$2NO_3^-$). The nitrate ion chemical ionization is a very selective method as nitrate ions react only with strong acids, such as malonic acid, sulfuric acid and methane sulfonic acid (Eisele and Tanner, 1993) and oxidized organic compounds that have at least two hydroperoxy (OOH) groups or other H-bond-donating groups (Hyttinen et al., 2015).

In the chemical ionization inlet ~20 liters per minute (lpm) of sheath flow is mixed with ~5 milliliters per minute flow of air saturated with nitric acid ($HNO_3$) and then guided to the ionization source. In the ionization source, nitric acid is ionized with



a soft x-ray source (Hamamatsu). The sheath flow with nitrate ions is guided to the drift tube where it gets mixed with the sample flow (10 lpm). The nitrate ions (or ion-clusters) and the molecules of the sample have around 200 ms of time to react with each other in drift tube before they enter the APi of the mass spectrometer through a 0.3 mm critical orifice with flow of
~0.8 lpm. In the APi the molecules of the sample are gradually pumped out while the ions are kept in the middle of the stream by quadrupoles and ion lenses (Junninen et al 2010). In the TOF the ions are accelerated with an energy pulse and they are separated by their time of flight to reach the detector in the chamber.

The CI-APi-TOF measurements were conducted at a 35-meter altitude in the same area as the ground level particle measurements. The measurement height is above the forest canopy. Zha et al. (2018) found that the HOM concentrations
above and inside the canopy are similar when the boundary layer is well-mixed. The concentrations between these altitudes may differ during a strong temperature inversion or a shallow surface layer in nighttime.

All the low-volatility vapor measurements were performed with the same instrument that was calibrated twice during this measurement period with sulfuric acid calibrator (Kürten et al., 2012). In the calibrations we achieved calibration coefficients $2.4e9$ $cm^{-3}$ for 2014-2018 and $4.6e9$ $cm^{-3}$ for 2019 onwards and used the same coefficient for all detected
compounds. This assumption is valid for compounds that cluster with nitrate ions at the collision limit and have equal collision rates. The collision rates of nitrate ions with SA and with HOMs are approximated to be very similar (Ehn et al., 2014). Mass spectra obtained from the instrument were analyzed using the "tofTools" program described in Junninen et al. (2010) and unit mass resolution was used in peak integration. The uncertainty of the concentrations is estimated to be -50%/+100%.


## 2.4 Complementary data

The NAIS measures the mobility distribution of ions in the atmosphere between 0.8 and 40 nm and it can be used to measure either naturally charged ions or the particles can be charged with a corona discharge to measure total particle concentration (Kulmala et al., 2007). We used an automatic atmospheric NPF event classification algorithm developed by Dada et al.
(2018) to determine NPF event times during the investigated time span. The event classification algorithm provided the start, peak and end times of NPF events using data from the NAIS. Relative humidity data was used from the Rotronic MP102H RH sensor in the measurement mast at the SMEAR II station, measured at 16 (before 2/2017) and 35-meter heights. The global radiation data was measured at the same measurement mast, with the Middleton SK08 pyranometer at 18-meter height (before 9/2019) and with the EQ09 pyranometer at 35-meter height.

## 2.5 Analysis methods for comparing PSM and CI-APi-TOF data

We used the time series of quality controlled and inverted sub-3nm particle concentrations to study the diurnal and seasonal patterns of sub-3nm aerosol particles. The same seasonal analysis was performed on the available CI-APi-ToF data.





Then, measured sub-3nm particle concentrations were then compared to the vapor concentrations to determine correlations between observed particle and vapor concentrations during NPF events. In order to ignore the effect of the diurnal cycles on the analysis, only events that occurred between 10:00 and 14:00 were included in the correlation analysis. Correlations were also separately investigated for spring- and summertime NPF events. There were not enough data points for events during autumn and winter for separate analysis during those seasons.

We compared the particle concentrations with measured SA and HOM concentrations since they have been identified to participate in NPF in laboratory studies (Sipilä et al., 2010; Kirkby et al., 2016). The HOM molecules were divided to monomers and dimers, as well as nitrates and non-nitrates according to their elemental composition. For each category, we summed up the concentrations of the selected peaks. For our purpose, it is not necessary to identify all possible peaks in each category, but to obtain the temporal variation of different types of particle precursor.

For SA, HOM non-nitrate monomers and HOM nitrate monomers, we chose mass peaks 97 Th ($HSO_4^-$), 340 Th ($C_{10}H_{14}O_9(NO_3^-)$) and 339 Th ($C_{10}H_{15}O_8N_2(NO_3^-)$), respectively (Sarnela et al., 2018; Kulmala et al., 2013). In order to increase the signal-to-noise ratio, we also chose extra mass peaks based on correlation and added up their signal. For SA, we chose all mass peaks for which the logarithmic correlation coefficient with the SA monomer was higher than 0.95, and for the organic molecules we chose mass peaks for which the logarithmic correlation with the original peak was higher than 0.85. Some of the selected peaks can contain several compounds including HOM nitrate monomers and radicals. For HOM dimers, we chose known peaks from previous studies because there was not one dominant peak to choose. For HOM non-nitrate dimers we selected 480 ($C_{18}H_{26}O_{11}(NO_3^-)$), 494 ($C_{19}H_{28}O_{11}(NO_3^-)$), 510 ($C_{20}H_{32}O_{11}(NO_3^-)$), 542 ($C_{20}H_{32}O_{13}(NO_3^-)$), 556 ($C_{20}H_{30}O_{14}(NO_3^-)$), 574 ($C_{20}H_{32}O_{15}(NO_3^-)$), 588 ($C_{20}H_{30}O_{16}(NO_3^-)$) and 620 Th ($C_{20}H_{30}O_{18}(NO_3^-)$) based on Sarnela et al. (2018). For HOM nitrate dimers we selected 538 ($C_{20}H_{32}O_{11}N_2(NO_3^-)$), 555 ($C_{20}H_{31}O_{13}N(NO_3^-)$), 570 ($C_{20}H_{32}O_{13}N_2(NO_3^-)$), 586 ($C_{20}H_{32}O_{14}N_2(NO_3^-)$) and 602 Th ($C_{20}H_{32}O_{15}N_2(NO_3^-)$) from known peaks (Zha et al., 2018). The complete list of peaks used in our analysis and their molecular formulas is listed in Table 2.

Sub-3nm particle concentrations were also compared to combinations of different precursor molecule concentrations since particle formation might involve several different vapor species. Laboratory experiments replicating boundary layer NPF in forested regions (Riccobono et al., 2014; Lehtipalo et al., 2018) and analysis of field data sets (Paasonen et al., 2010) have shown that particle formation rates can be parametrized using a product of sulfuric acid concentration and organics concentrations.

**Table 2: The mass peaks selected for analysis and their molecular formulas. Sulfuric acid and HOM monomer peaks were selected based on correlation with the bolded mass peaks and summed together in order to increase the signal-**



to-noise ratio. HOM nitrate monomer peaks listed as "Several compounds" contain HOM nitrate monomers and radicals, but a single peak cannot necessarily be identified as the main compound.

| Sulfuric Acid | | HOM Non-nitrate Dimers | |
|---|---|---|---|
| **97 Th** | $HSO_4-$ | 480 Th | $C_{18}H_{26}O_{11}(NO_3^-)$ |
| 160 Th | $H_2SO_4(NO_3^-)$ | 494 Th | $C_{19}H_{28}O_{11}(NO_3^-)$ |
| 195 Th | $H_2SO_4 \cdot HSO_4^-$ | 510 Th | $C_{20}H_{32}O_{11}(NO_3^-)$ |
| HOM Non-nitrate Monomers | | 542 Th | $C_{20}H_{32}O_{13}(NO_3^-)$ |
| 298 Th | $C_8H_{12}O_8(NO_3-)$ | 556 Th | $C_{20}H_{30}O_{14}(NO_3^-)$ |
| 308 Th | $C_{10}H_{14}O_7(NO_3^-)$ | 574 Th | $C_{20}H_{32}O_{15}(NO_3^-)$ |
| 310 Th | $C_{10}H_{16}O_8(NO_3^-)$ | 588 Th | $C_{20}H_{30}O_{16}(NO_3^-)$ |
| **340 Th** | $C_{10}H_{14}O_9(NO3^-)$ | 620 Th | $C_{20}H_{30}O_{18}(NO_3^-)$ |
| HOM Nitrate Monomers | | HOM Nitrate Dimers | |
| 297 Th | Several compounds | 538 Th | $C_{20}H_{32}O_{11}N_2(NO_3^-)$ |
| 311 Th | Several compounds | 555 Th | $C_{20}H_{31}O_{13}N(NO_3^-)$ |
| 327 Th | Several compounds | 570 Th | $C_{20}H_{32}O_{13}N_2(NO_3^-)$ |
| **339 Th** | $C_{10}H_{15}O_8N_2(NO_3^-)$ | 586 Th | $C_{20}H_{32}O_{14}N_2(NO_3^-)$ |
| | | 602 Th | $C_{20}H_{32}O_{15}N_2(NO_3^-)$ |

## 3 Results

In the following section we present the 74-month time series of sub-3nm particle concentrations and the 31-month time series of aerosol precursor vapors measured at the SMEAR II –station in Hyytiälä, southern Finland and their comparison for the overlapping time period.

### 3.1 Time series of particle concentrations

The entire time series of the particle concentrations are shown in Fig. 4. The concentrations show a clear seasonal pattern for all three size bins: 1.1-1.3 nm, 1.3-1.7 nm and 1.7-2.5 nm. We observe a clear annual maximum during late spring and early summer, and we also observe the lowest concentrations during the winter months, consistent with earlier observations at the same site (Kontkanen et al., 2017). The end of year 2016 and early 2017 show lower total concentrations than the rest of the time series. This could be due to atypical environmental conditions, but we cannot exclude technical reasons leading to diminished detection efficiency (the instrument was thereafter changed in spring 2017). Excluding this part of the data did

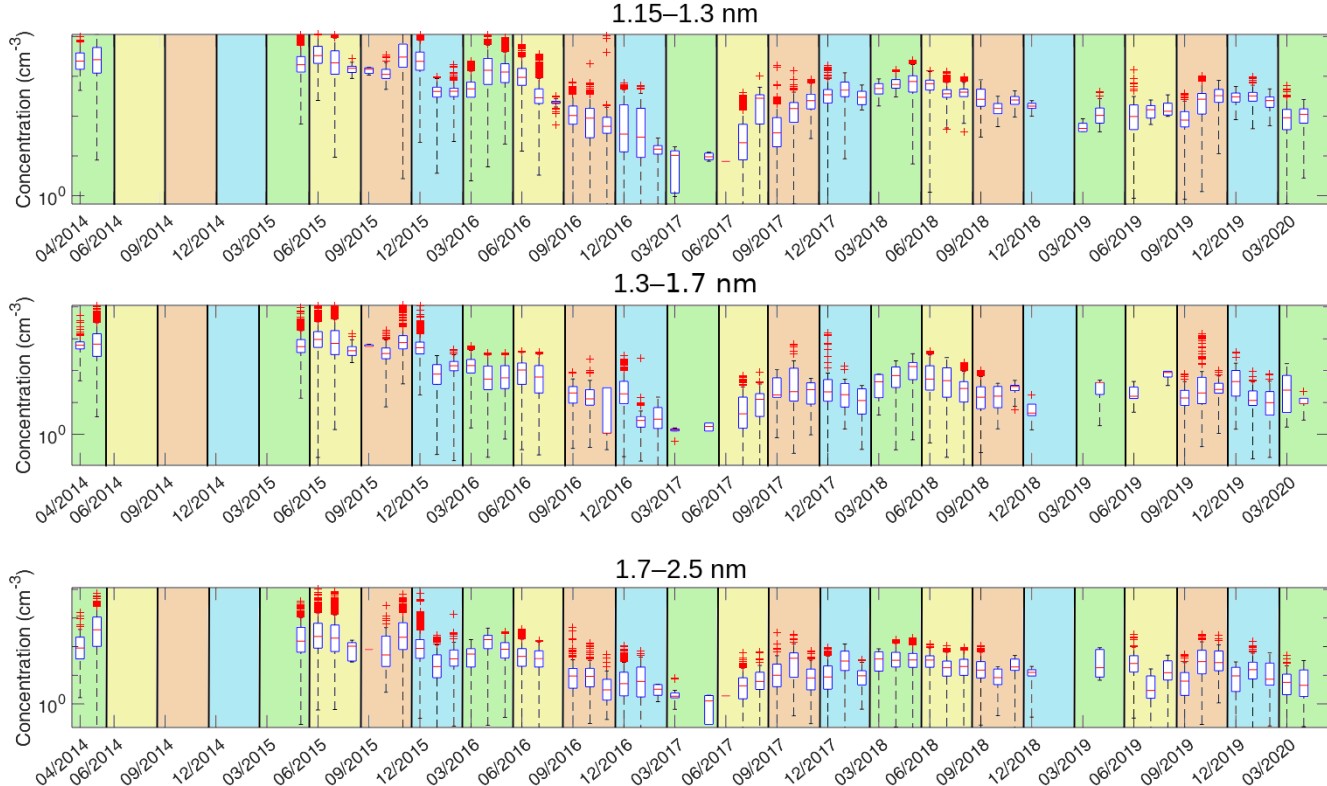


**Figure 4: The monthly variation of the particle concentration in the three PSM size bins. The red line shows the median concentration for each month and the blue box contains 50% (25th to 75th percentile) of all data points. The whiskers mark the location of the 95th and 5th percentile data points and the red plusses are outliers beyond those percentiles. The areas with the green background are spring months, the yellow background represents summer months, the brown background represents autumn months and the blue background represents winter months.**


not have a significant impact on the rest of the analysis. The median sub-3nm concentration of the entire dataset from 2014 to 2020 was $4.1 \times 10^2$ cm$^{-3}$, with the spring and summertime concentration being $5.3 \times 10^2$ cm$^{-3}$ and the autumn and wintertime concentration being $3.3 \times 10^2$ cm$^{-3}$ concentration. Kontkanen et al. (2017) ended up with somewhat higher

concentrations (median $2.0 \times 10^3$ cm$^{-3}$), although the variation in concentrations is similar. That is likely explained by the slightly wider size range in that study (up to 3 nm, where the largest size bin was obtained from the difference between PSM and a differential mobility particle sizer) and because their data was not filtered to remove scans with too high background. The two months of measurements from 2014 at the beginning of the time series show a higher median concentration than the rest of the data, which could be due to the difference in background removal, as the background was measured manually at

that time.



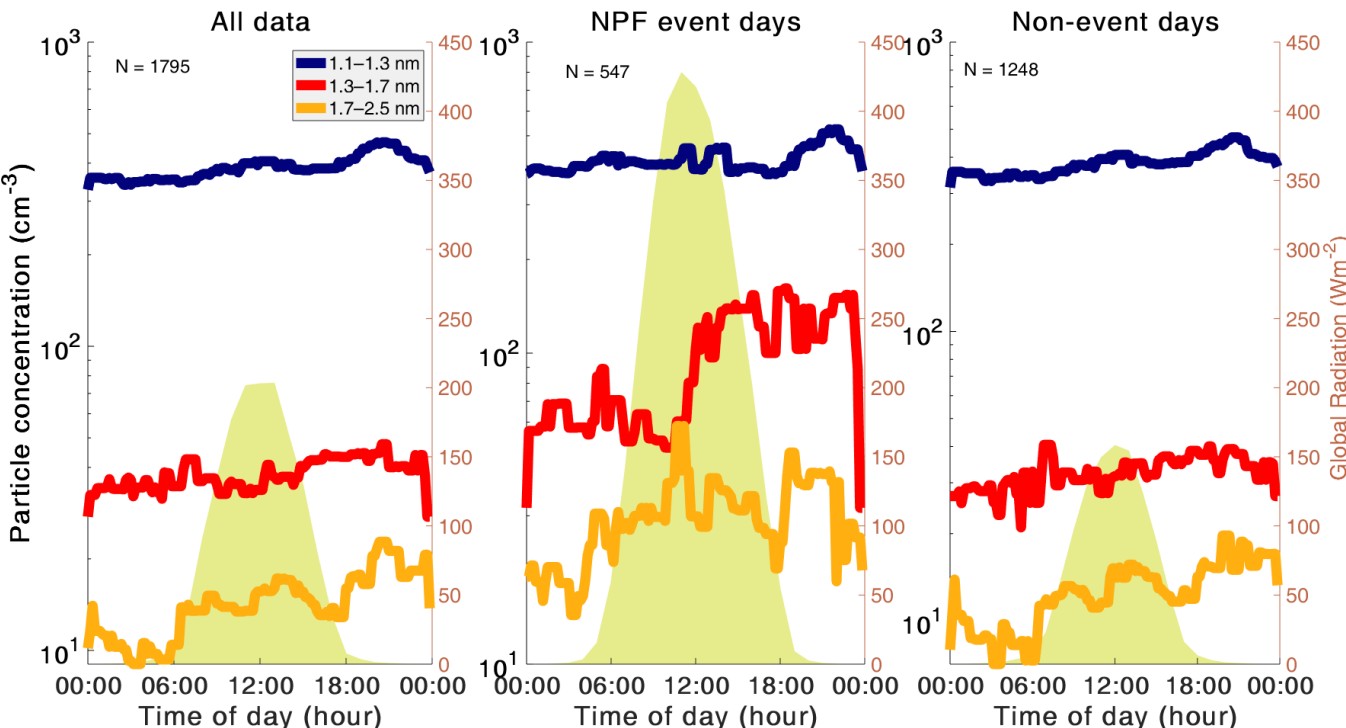

**Figure 5: The median diurnal cycles of the particle concentrations for the entire dataset (left panel), new particle formation event days (middle panel) and non-event days (right panel). The blue, red, and yellow lines show the concentrations for size ranges of 1.1-1.3 nm, 1.3-1.7 nm, and 1.7-2.5 nm, respectively. The number of days included in the median day is presented by the N at the top left of each figure. The light green shading is the diurnal cycle of global radiation (right y-axis).**

The diurnal patterns of particle concentrations in three size bins are shown in Fig. 5. We observe two maxima for the 1.1–1.3 nm concentration: one around midday and another during the evening. In this size bin, the measured concentrations can consist of both very small particles, large gas molecules or molecular clusters (Ehn et al., 2014); the distinction between them cannot be made based on the measurement. Consequently, the daytime maximum can result from a combined effect of the diurnal behaviors of large organic molecules and newly formed molecular clusters. The diurnal variation of organic compounds is discussed below. Similarly, the evening-time maximum can be due to organic molecules or molecular clusters, which have been observed to form during evening time by biogenic ion-induced mechanism (Rose et al., 2018). Both 1.3–1.7 nm and 1.7–2.5 nm particle concentrations exhibit a daytime maximum in the afternoon. However, we observe a second, larger maximum for the 1.7–2.5 nm particle concentration in the evening as well.

During regional NPF events, we expect the sub-3nm size distribution to behave differently than when there is no event, in the case that the formation of small particles takes place at our measurement location. In Fig. 5, we present diurnal cycles for NPF event and non-event days separately. Even though the event classification algorithm gives us exact event times, we used





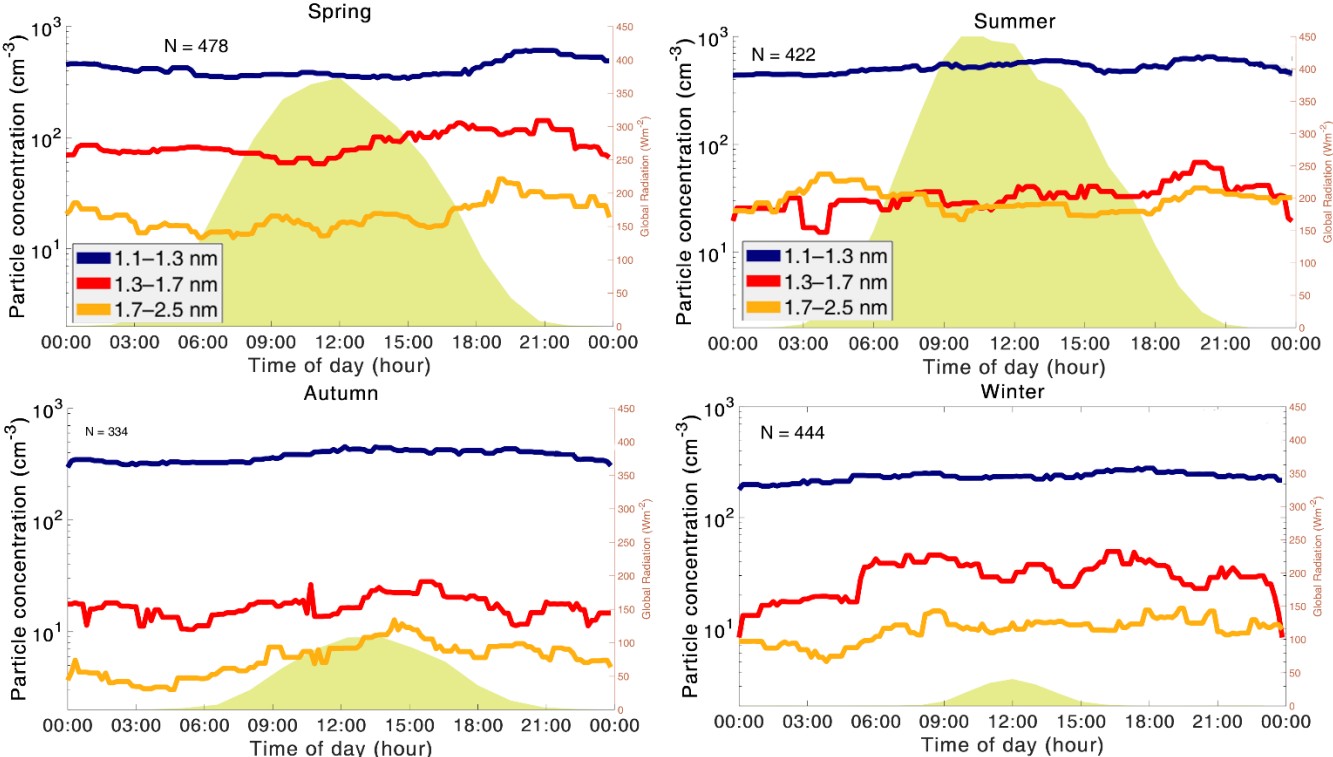

**Figure 6: The median diurnal cycles of the particle concentrations in different seasons. At the top left is the diurnal cycle for spring, top right is for summer, bottom left for autumn and bottom right for winter. The yellow, red, and blue lines show the concentrations for the size ranges of 1.15-1.3 nm, 1.3-1.7 nm, and 1.7-2.5 nm, respectively. The number of days included in the median day is presented by the N at the bottom right of each figure. The light green shading is the diurnal cycle of global radiation (right y-axis).**

entire event days in this part of the analysis. The most noticeable difference between NPF event and non-event days is the strong midday maximum for both 1.3–1.7 nm and 1.7–2.5 nm particle concentrations on NPF event days. This maximum does not appear during non-event days, leading to the conclusion that the increase in midday concentrations can be attributed to regional NPF. The concentrations are also generally higher during NPF than non-NPF days, indicating that conditions are favorable for cluster/particle formation. However, the smallest size bin does not show a clear difference during NPF and non-NPF days. This could mean that the production and sinks of 1.1–1.3 particles are large enough that the enhanced growth into the 1.3–1.7 particle size range during event times is not visible in the concentration. It also confirms that there is a constant concentration of small particles/clusters present in the atmosphere, much like ion clusters (Kulmala et al. 2007; 2013; Kontkanen et al. 2017).

Because the formation mechanism and thus particle concentrations can vary depending on the season, we studied the diurnal cycles of each particle concentration size class separately for each season (Figure 6). During springtime, the diurnal pattern

of 1.1–1.3 nm concentration has two maxima, a daytime, and an evening maximum. We observe up to three times larger concentration of 1.3–1.7 nm particles during spring compared to the average diurnal concentrations for other seasons. This increase in particle concentration likely results from the increased frequency of regional NPF during spring (Nieminen et al., 2014).

During the summer, we observe the highest 1.1–1.3 nm particle concentrations out of all seasons. This is likely due to the increased biogenic activity in the surrounding forest and stronger photochemical production of precursor vapors. The diurnal pattern of the 1.1–1.3 nm concentration exhibits the same two maxima as the diurnal pattern of the entire dataset. However, the 1.3–1.7 nm particle concentration does not show a clear diurnal pattern, and the 1.7–2.5 nm concentration only exhibits a weak early morning maximum and a minimum between 16:00 and 17:00. This lack of diurnal pattern and the low

concentration of 1.3–1.7 nm and 1.7–2.5 nm particle concentration is likely linked to a decreased frequency of NPF during summer (Dada et al., 2017).

In autumn, the 1.1–1.3 nm particle concentration has a weak but otherwise similar diurnal pattern to summer- and springtime concentrations. We observe no discernable diurnal pattern for the 1.3–1.7 particle concentration. The 1.7–2.5 nm particle

concentration has a weak diurnal pattern, but the overall concentration is notably lower than in summer and spring, particularly during night-time.

During the winter, we observe the lowest median concentrations of 1.1–1.3 nm particle concentrations. Additionally, the concentrations of 1.1–1.3 nm, 1.3–1.7 nm and 1.7–2.5 nm particle concentrations exhibit only a single, small midday

maximum. This supports the idea that the evening maximum is related to the formation of organic clusters (Rose et al., 2018). The fact that sub-3nm particle concentrations during winter are generally low is consistent with earlier observations that NPF is rare during wintertime (Dada et al. 2017; Nieminen et al. 2014).

## 3.2 Time series of vapor concentrations

The time series of the precursor vapors are shown in Fig. 7. We observe the highest concentrations of all studied vapors during summer months and the lowest concentration in either late autumn, winter, or early spring. There are no visible differences in concentrations between different years, but there are more outliers in the vapor concentrations during 2014 and



2015.

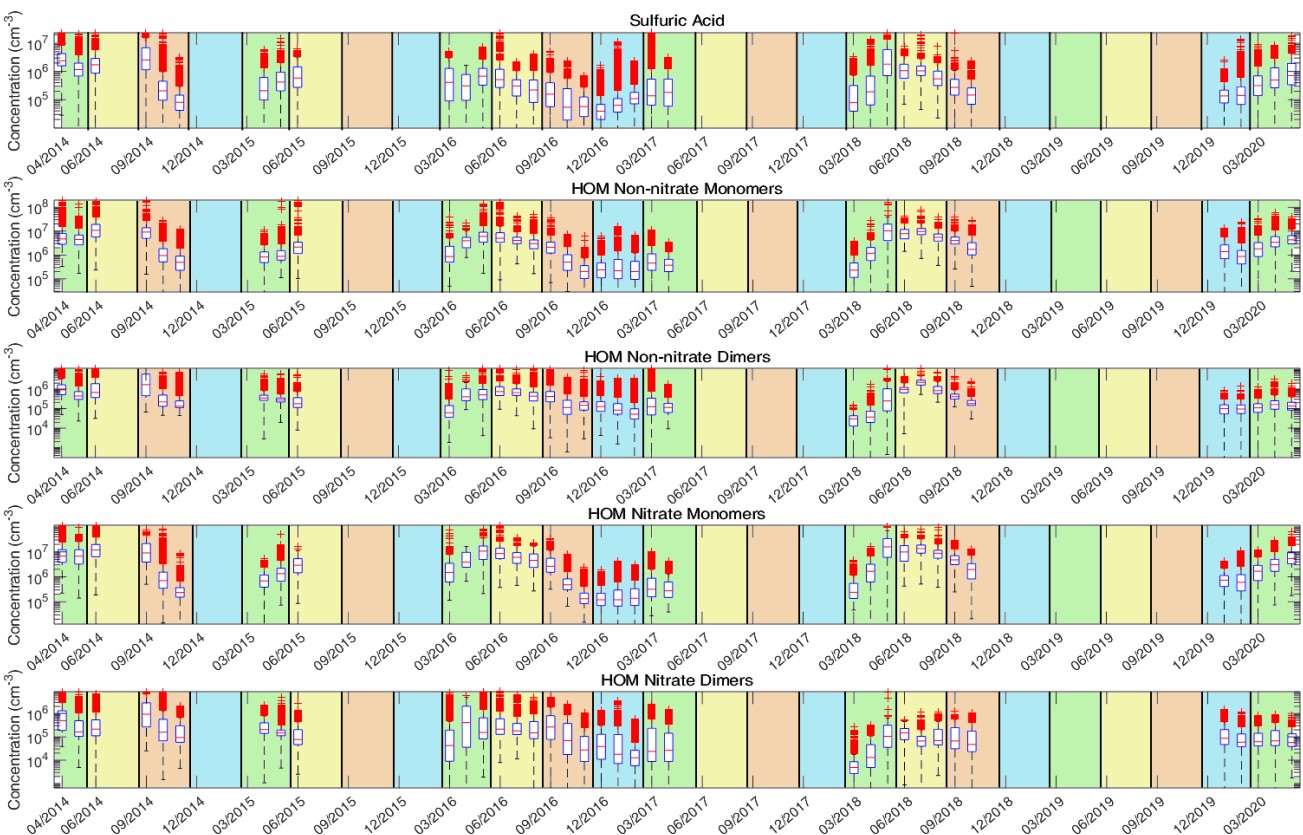

**Figure 7: The time series of the selected atmospheric aerosol precursor vapors. From the top down, the time series are shown for sulfuric acid, HOM non-nitrate monomer, HOM non-nitrate dimer, HOM nitrate monomer and HOM nitrate dimer concentrations, respectively. The time series are from April 2014 to April 2020. The red line is the median concentration for each month and the blue box contains 75% of all data points. The whiskers mark the location of the 95th and 5th percentile data points and the red plusses are outliers beyond those percentiles. The areas with the green background are spring months, the yellow background represents summer months, the brown background represents autumn months and the blue background represents winter months.**

We also analyzed the diurnal behavior of the measured aerosol precursor vapors in the same fashion as the particle concentrations discussed above. The diurnal patterns of SA, HOM monomer (nitrate and non-nitrate) and HOM dimer (nitrate and non-nitrate) concentrations for the entire dataset are shown in Fig. 8. SA concentration has a similar diurnal pattern to that of global radiation, which is expected as sulfuric acid is formed in the atmosphere mainly through photochemical oxidation (Lucas and Prinn, 2005; Petäjä et al., 2009). HOM non-nitrate monomer concentration has a minimum in the early morning, with the concentration rising throughout the day until the maximum is reached after 18. In



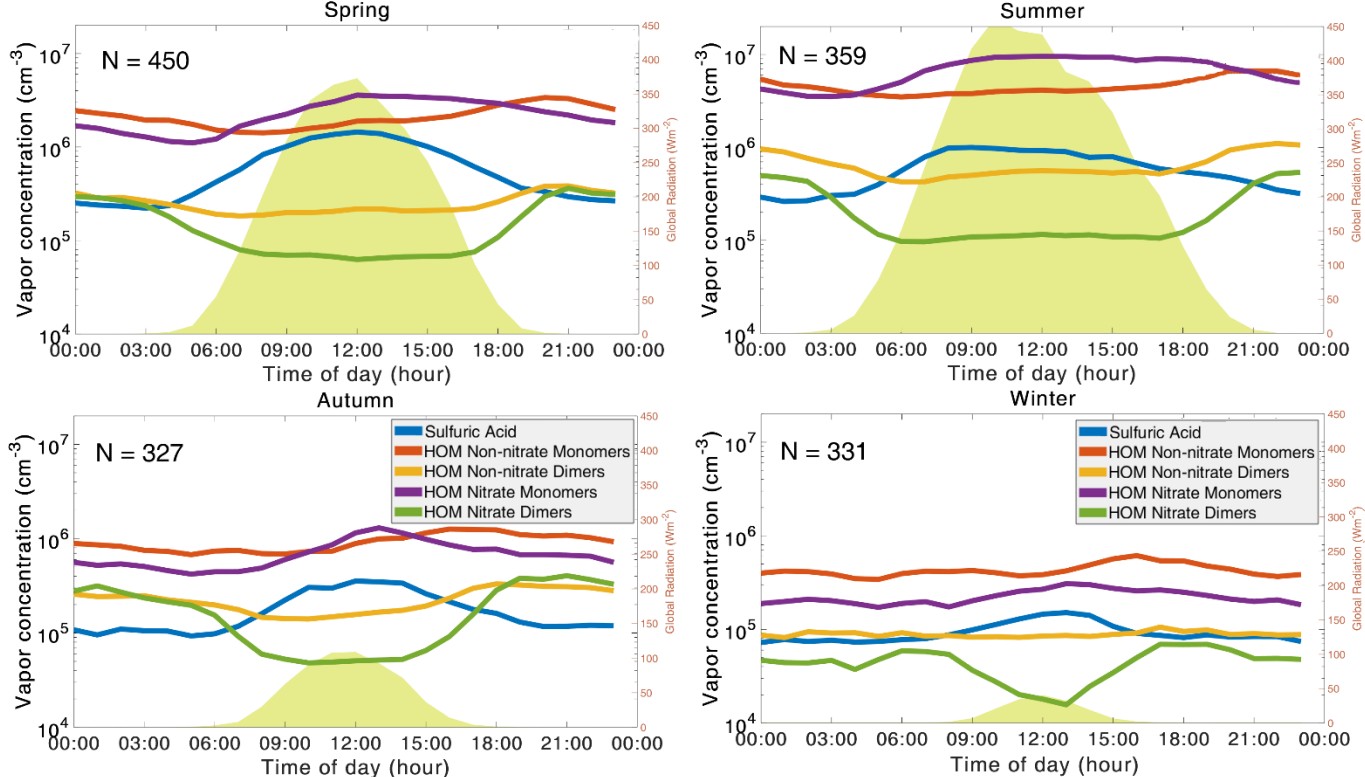

**Figure 9: The diurnal cycles of precursor vapors in different seasons. At the top left there is the diurnal cycle in spring, at the top right there is the diurnal cycle in summer, at the bottom left there is the diurnal cycle in autumn and at the bottom right there is the diurnal cycle in winter. The blue line is the sulfuric acid concentration, red line is the HOM non-nitrate monomer concentration, the yellow line is the HOM nitrate monomer concentration, the yellow line is the HOM non-nitrate dimer concentration and the green line is the HOM nitrate dimer concentration. The number of days included in the median day is presented by the N in each figure. The light green shading is the diurnal cycle of global radiation (right y- axis).**

contrast, HOM nitrate monomer concentration exhibits a single daytime peak around midday similar to the sulfuric acid concentration. HOM dimer (both nitrate and non-nitrate) concentrations have different diurnal cycles than the other vapors, exhibiting minima during daytime and an increased concentration at night. Similar patterns were found by Bianchi et al. (2017) using CI-APi-ToF data from spring 2013 in Hyytiälä and Jokinen et al. (2017) during a solar eclipse.

During regional NPF event days, the concentrations of all analyzed aerosol precursor vapors are higher than during non-event days (Figure 8). However, the diurnal patterns of the precursor vapors are otherwise similar on event and non-event days. Additionally, we observe that the aerosol precursor vapor concentrations rise earlier and the difference in concentrations between the night-time and the daytime is larger on NPF event days. These observations suggest that during

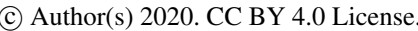



event days there is more photochemical production and potentially also higher emissions of biogenic vapors. This is further corroborated by the diurnal patterns of global radiation in Figures 5 and 8.

The diurnal behavior of SA has a similar daytime maximum as the 1.1–1.3 nm particle concentration when we compare the diurnal behaviors of the entire particle and vapor concentration data sets. Additionally, the diurnal behavior of sulfuric acid matches that of the 1.3–1.7 nm and 1.7–2.5 nm particle concentrations during event days. This points to the importance of sulfuric acid in atmospheric cluster formation and in the initial stages of aerosol growth. The HOM non-nitrate monomer maximum in the evening coincides with the observed peak in 1.1–1.3 nm particle concentration, implying that this maximum

is likely due to formation of organic clusters.

In Fig. 9, we show the diurnal patterns of precursor vapor concentrations separately for each season. During spring, the diurnal patterns of all studied vapors exhibit similar behavior when compared to the diurnal behavior in the entire dataset. This underlines the effect of spring- and summertime data on the entire dataset because the diurnal cycles during these

seasons are fairly similar and will therefore dominate the dataset.

In summer, the diurnal behaviors of sulfuric acid and HOM dimer concentrations are fairly similar to springtime behavior. The HOM monomer concentration does not rise as strongly during the day as during spring, but we still observe a sharp increase in HOM non-nitrate monomer concentration during the evening. Also, in summertime HOM nitrate monomer

concentration reaches a maximum earlier than in spring, around 9:00 This is most likely because there is solar radiation available for a longer period of time during summer. Overall, the HOM monomer concentrations are as much as five times higher during the summer than during spring and the concentrations of other HOMs are higher as well. The observed high HOM concentrations during summer can be explained by high emissions of organic vapors from the surrounding forest (Hellén et al., 2018) and increased photochemical activity.


In autumn, sulfuric acid concentration shows a daytime maximum around midday, but the concentration rises later in the morning and decreases earlier in the afternoon than during spring and summer. Additionally, the daytime sulfuric acid peak is smaller. Similarly, HOM monomer concentrations begin to rise later during the morning than in spring or summer, most likely because of the seasonality of solar radiation's diurnal behavior. Overall, the median concentrations and diurnal

behaviors during autumn are comparable to springtime diurnal behaviors of the vapors, with the exception that the concentration of HOM nitrate monomers is lower and more comparable to winter concentrations. HOM dimer diurnal behaviors are similar to spring, but nighttime concentrations are higher and the period of decreased concentrations during the day is shorter. These are again likely due to the seasonal differences in global radiation.





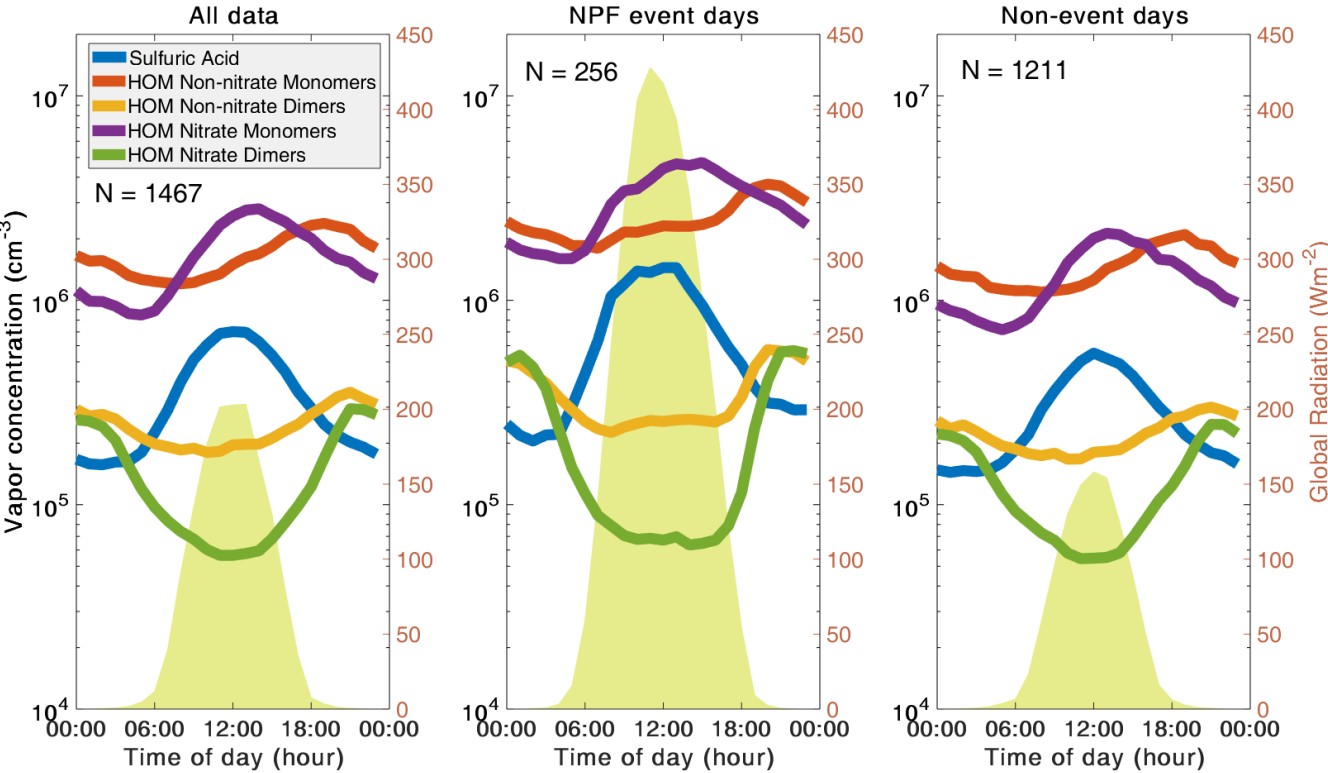

**Figure 8: The diurnal cycle for each selected atmospheric precursor vapor for the entire dataset (left panel), for new particle formation event days (middle panel) and for non-event days (right panel). The blue line is the sulfuric acid concentration, red line is the HOM non-nitrate monomer concentration, the purple line is the HOM nitrate monomer concentration, the yellow line is the HOM non-nitrate dimer concentration and the green line is the HOM nitrate dimer concentration. The number of days included in the median day is presented by N at the top right of each figure. The light green shading is the diurnal cycle of global radiation (right y-axis).**

In winter, the concentrations of precursor vapors are lower than during other seasons and their diurnal variation is smaller. Sulfuric acid and HOM monomer concentrations begin to rise later in the morning, around 09:00-10:00, again likely due to the seasonality of solar radiation diurnal behavior. HOM nitrate dimer concentration behavior remains similar to other seasons, but the median concentration is lower. Notably, the HOM non-nitrate dimer concentration has a barely detectable diurnal cycle during the winter.


### 3.3 The connection between precursor vapors and the formation of sub-3nm particles

We used correlation analysis to investigate the relationship between atmospheric sub 3nm-particle concentrations and the selected atmospheric vapors. The data is from NPF event times as specified by the NPF event algorithm and only from events occurring between 10 and 14 as to diminish the effect of the diurnal cycles on the correlations. Limiting our data selection to this time range diminishes the effect of meteorological variables on our analysis and allows us to focus on daytime NPF events. Because absolute vapor concentrations are not needed for this analysis, we used the aerosol precursor

vapor data without the calculated calibration coefficients to eliminate this source of uncertainty. The results are shown in Table 2.

Particle concentrations during NPF events show clear correlations with sulfuric acid and with HOM dimer concentrations. Notably, sulfuric acid concentration correlates particularly with 1.7–2.5 nm concentrations, underlining its importance in the

initial growth of the newly formed particles or determining when NPF events happen. The smallest particles correlate better with HOMs than SA, especially when looking at the whole data set, confirming that this size range is influenced by organic molecules or clusters. HOM monomers correlate only with the smallest size bin, while HOM dimers correlate with all sub-3nm concentrations measured with the PSM, especially during event times. This indicates that HOM dimers are important in the formation of the smallest atmospheric particles, a finding consistent with Lehtipalo et al. (2018). The best correlation is

found between HOM nitrate dimer concentration and 1.3–1.7 nm particle concentrations.

The combination of HOM (nitrate or non-nitrate) dimer and sulfuric acid concentration correlates most strongly with the 1.3–1.7 nm and 1.7–2.5 nm particle concentrations. This is consistent with laboratory experiments in the

CLOUD chamber, showing that particle formation rates at 1.7 nm correlate with the product of sulfuric acid, ammonia and HOM dimers (Lehtipalo et al., 2018). However, we observe that HOM nitrate dimers have a better correlation with the particle concentrations than the HOM non-nitrate dimers. We would expect the opposite, as non-nitrate HOMs have lower volatility than nitrate HOMs (Yan et al., 2020). This discrepancy with the laboratory results could be explained by the nitrate HOMs being better correlated with global radiation (Figure 8), as NPF most frequently occurs at the SMEAR II station

during the global radiation maximum. However, the difference in correlation coefficients is not large, so it could also mean that at least some of the nitrate dimers already have low enough volatility to participate in NPF, especially together with SA.

The scatter plots for the best correlations between atmospheric vapors and sub-3nm concentrations are shown in Fig. 10. It is possible that both sulfuric acid and large organic molecules are required for the formation and growth of new particles,

which would explain the observed correlations. However, the correlation can also point to two separate formation pathways, organic and inorganic. HOM nitrate monomers, on the other hand, do not correlate with sub-3nm concentrations, although





**Table 2: Logarithmic Pearson correlations between particle concentrations in different size bins measured with the PSM and known precursor vapors or combinations of precursor vapors. The data from NPF events occurring between 10 and 14 is used.**
**Statistically significant values (p < 0.05) are bolded and values higher than 0.5 are highlighted with red color.**

|  | 1.1 – 1.3 nm concentration | | 1.3 – 1.7 nm concentration | | 1.7 – 2.5 nm concentration | |
|---|---|---|---|---|---|---|
|  | Events | All data | Events | All data | Events | All data |
| Sulfuric Acid (SA) | **0.27** | **0.37** | **0.37** | **0.3** | **0.57** | **0.37** |
| HOM Non-nitrate Monomers | **0.16** | **0.53** | 0.08 | 0.08 | 0.13 | 0.29 |
| HOM Non-Nitrate Dimers | **0.28** | **0.55** | **0.54** | **-0.04** | 0.26 | 0.25 |
| HOM Nitrate Monomers | 0.05 | **0.48** | -0.07 | **0.04** | 0.04 | **0.3** |
| HOM Nitrate Dimers | **0.53** | **0.49** | **0.62** | **0.12** | **0.52** | **0.33** |
| SA X HOM Nitrate Monomers | 0.08 | **0.45** | 0.24 | 0.14 | 0.39 | 0.32 |
| HOM Non-nitrate Monomers X HOM Nitrate Monomers | 0.11 | 0.5 | 0 | 0.06 | 0.09 | **0.3** |
| SA X HOM Non-nitrate Monomers | **0.19** | **0.47** | 0.36 | 0.18 | **0.5** | 0.34 |
| SA X HOM Non-nitrate Dimers | 0.33 | **0.52** | **0.62** | 0.18 | **0.61** | 0.4 |
| SA X HOM Nitrate Dimers | 0.48 | 0.43 | **0.77** | 0.31 | **0.66** | 0.44 |
| SA X HOM Non-nitrate Monomers^2 | 0.15 | 0.5 | 0.31 | 0.11 | 0.41 | 0.31 |
| SA^2 X HOM Non-nitrate Monomers^2 | 0.23 | 0.44 | 0.37 | 0.23 | **0.54** | 0.35 |
| SA^2 X HOM Non-nitrate Dimers | 0.32 | 0.49 | **0.53** | 0.25 | **0.62** | 0.41 |
| SA^2 X HOM Nitrate Dimers | 0.41 | 0.38 | **0.73** | 0.33 | **0.64** | 0.43 |
| SA^2 X HOM Nitrate Monomers | 0.15 | 0.43 | 0.3 | 0.2 | 0.47 | 0.35 |
| SA^2 | **0.27** | **0.37** | **0.37** | **0.3** | **0.57** | **0.37** |

they also show a daytime maximum like sulfuric acid, and in some earlier studies they have been connected to cluster formation (Jokinen et al., 2017). This supports the concept that HOM nitrate monomers have higher volatilities than other
HOMs, so they might participate in later growth of particles, but not in clustering and initial growth (Yan et al., 2020; Lehtipalo et al., 2018).





**Figure 10: The best correlations between particle concentrations measured by PSM in three size bins and precursor vapors during NPF events occurring between 10 and 14. The correlations of sub-3nm particle concentrations and sulfuric acid are on the left, the correlations of sub-3nm particle concentrations and HOM nitrate dimers are in the middle and the correlations of sub-3nm particle concentrations and the product of sulfuric acid and HOM nitrate dimers are on the right.**

Due to the differences observed in the diurnal cycles of both sub-3nm particle concentrations and vapor concentrations between different seasons, we investigated the correlation between sub-3nm particle concentrations and atmospheric vapors in different seasons. Because of the lack of data for both vapor and particle concentrations during winter NPF events, we





were only able to analyze spring, summer, and autumn events. The results of the seasonal NPF analysis are shown in Table 3.


**Table 3: Logarithmic correlations of known precursor vapors with particle concentrations in three size bins for spring (Spr.), summer (Sum.), and autumn (Aut.) seasons during event times between 10 and 14. Winter did not have enough available event points for correlation analysis. Statistically significant values (p < 0.05) are bolded and values higher than 0.5 are highlighted with red color.**

| | 1.1–1.3 nm concentration | | | 1.3–1.7 nm concentration | | | 1.7–2.5 nm concentration | | |
|---|---|---|---|---|---|---|---|---|---|
| | Spr. | Sum. | Aut. | Spr. | Sum. | Aut. | Spr. | Sum. | Aut. |
| Sulfuric Acid | 0.15 | -0.36 | 0.02 | 0.27 | 0.21 | 0.17 | **0.48** | 0.34 | -0.12 |
| HOM Non-nitrate Monomers | **-0.46** | 0.15 | **0.47** | **0.34** | -0.26 | 0.16 | -0.21 | **0.69** | **0.44** |
| HOM Non-nitrate Dimers | 0.19 | **-0.58** | **0.57** | **0.87** | 0.42 | 0.18 | **0.29** | -0.14 | **0.46** |
| HOM Nitrate Monomers | **-0.51** | 0.12 | **0.4** | 0.18 | -0.23 | 0.16 | -0.06 | **0.56** | 0.25 |
| HOM Nitrate Dimers | **0.61** | 0.31 | **0.54** | **0.74** | -0.06 | -0.04 | **0.45** | 0.13 | **0.64** |


The analysis reveals clear seasonal differences between correlations of precursor vapors and sub-3nm particle concentrations. During springtime NPF events, HOM dimers correlate with all size bins of measured sub-3nm particle concentrations. This is consistent with the results of Yan et al. (2018), who compared vapor concentrations with particle measurements performed with the NAIS. Additionally, sulfuric acid correlates well with 1.7–2.5 nm particle concentrations.

Interestingly, HOM monomers anticorrelate with 1.1–1.3 nm and 1.7–2.5 nm particle concentrations during NPF events in spring, maybe because dimer formation is a sink of the monomers.

During summertime events, we do not observe as many statistically significant correlations. HOM monomers correlate with 1.7–2.5 nm particle concentrations while correlations HOM non-nitrate dimers anticorrelate with 1.1–1.3 nm particle concentrations. It is possible that this is caused by the higher evaporation rate of HOM dimers with the increased temperature

(Donahue et al.,2011). However, due to the lack of vapor concentration data from summer months, the amount of data available for analysis here is limited and thus correlations may not be representative.

During autumn, HOM monomers correlate with the 1.1–1.3 nm particle concentration, a notable difference from spring. HOM non-nitrate monomers also correlate with the 1.7–2.5 nm particle concentration. In addition, both HOM dimers correlate with 1.1–1.3 nm and 1.7–2.5 nm particle concentrations. We also do not observe statistically significant

correlations with the 1.3 – 1.7 nm particle concentration and precursor vapors. These differences in correlations between particle and vapor concentrations point to an annual variation in the formation mechanisms of sub-3nm particles.

It should be noted that SA do not form particles by itself at concentrations relevant to atmospheric boundary layer (Kirkby et al. 2011). Rather, it needs ammonia (NH$_3$) or amines to stabilize the forming clusters. It is yet unclear if SA can form stable
clusters with HOMs, although SA-organics nucleation has been proposed (e.g. Riccobono et al. 2014). Lehtipalo et al. (2018), showed that SA and HOMs do not to interact unless NH$_3$ is present. As there is no continuous ammonia and amine measurements available at SMEAR II, we could not include those in the correlation analysis, although variations in these vapors can affect the NPF mechanism and thus our results, especially the seasonal variation. Hemmilä et al. (2018) showed that there is a weak positive correlation between 1-2 nm particles measured with the PSM and ammonia and dimethylamine
concentrations.

## 4 Conclusions

In this study, we analyzed five years of sub-3nm particle concentration and aerosol precursor vapor concentration data from the SMEAR II station in Hyytiälä, southern Finland. The sub-3nm particle concentrations were measured with the PSM and
the aerosol precursor molecule concentration data was measured with the CI-APi-ToF.

The analysis of the PSM background counts and stability shows that to operate the PSM at the SMEAR II station in such a way that it reliably activates sub-3nm atmospheric particles, the measured background in the PSM should be within 1 and 10 cm$^{-3}$. Too low a background, and consequently too low a supersaturation level in the PSM, results in poor activation of
atmospheric sub-3nm particles. When the supersaturation is too high, the measurement becomes unstable and the observed concentrations are affected by homogenic nucleation of the working fluid. The settings of the PSM indicated by this analysis are valid for the SMEAR II station and other similar boreal background stations, but when measuring in other environments, the optimal background level may be different.

The size distribution of sub-3nm particles shows a clear seasonal cycle. The 1.1–1.3 nm particle concentrations have the highest concentrations during the summer, which coincides with increased summertime photochemical activity and biogenic activity in the surrounding forest. The 1.3–1.7 nm and 1.7–2.5 nm particle concentrations show a marked increase during springtime, coinciding with increased regional NPF frequency. The diurnal patterns of sub-3nm concentrations exhibit clear daytime maxima around midday. This maximum is the clearest during spring and autumn, during which regional NPF events
are also most common. A second maximum in the evening is observed for the 1.1–1.3 nm particles during spring and



summer, but not for the concentrations in bigger size ranges or during wintertime. This maximum may be linked to organic clusters that form but do not grow to larger particles in the atmosphere.

The precursor vapors also show seasonal variability. The concentrations of all selected precursor vapors are the highest during summer and the lowest during winter. This is attributed to increased biogenic activity in the surrounding forest during the warmer periods of the year as well as increased photochemical production. Additionally, the concentrations of sulfuric acid and HOM monomers have seasonally changing diurnal behavior because of solar radiation.

When comparing sub-3nm particle concentrations with aerosol precursor vapors, we found that the smallest particles (1.1-1.3 nm) correlate with HOMs when looking at the whole time series, indicating their presence in this size range. The 1.3-1.7 nm and 1.7-2.5 nm particles, which are more directly connected to NPF events, correlate with SA and HOM dimers (and the product of these) during NPF events, but not with HOM monomers. There was no significant difference between nitrate and non-nitrate HOMs regarding their correlations with sub-3nm particles. The seasonal analysis of the correlations reveals some differences between the seasons, which could be due to changes in the mechanism forming clusters. However, 610 understanding the seasonal differences in the formation mechanisms of HOMs and sub-3nm particles in detail requires further studies.

*Data availability.* The meteorological data from the SMEAR II station can be accessed from the smartSMEAR website: http://avaa.tdata.fi/web/smart/. The data is licensed under a Creative Commons 4.0 Attribution (CC BY) license. The particle 615 and vapor concentration data measured with PSM, CI-APi-ToF and NAIS are available from the authors upon request.

*Author contribution.* JS, NS, LA, TL and TJ conducted the measurements, JS and NS handled the data inversion and JS performed the analysis and wrote the paper. All of the authors contributed to the discussions of the results and commenting on the paper.


*Competing interests*. The authors declare that they have no conflict of interest.

*Acknowledgements.* This work was supported by the Academy of Finland (grant nos. 316114, 307331, 311932, 1325656), the Academy of Finland via Centre of Excellence in Atmospheric Sciences (272041), the University of Helsinki (grant no. 625 75284132) and the European Union's Horizon 2020 research and innovation program under grant agreements No.654109 and 739530 (ACTRIS). SMEAR II staff is acknowledged for their help in running the measurements.





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
