# Peer review of "LONG-TERM MEASUREMENT OF SUB-3NM PARTICLES AND THEIR PRECURSOR GASES IN THE BOREAL FOREST"

_Atmospheric Chemistry and Physics, 2020_

## Referee Comment (RC1) · Anonymous Referee #1 · 5 Sep 2020

In this manuscript, five years of sub-3 nm particle concentrations obtained with the PSM and three years of precursor vapor concentrations measured with the CI-APi-ToF at the boreal environment are presented, and possible correlation between particles and vapors are examined. This is a unique dataset analyzed and the results are definitely worth being presented in ACP after few minor revisions.

A general comment is the presentation of variabilities of the various atmospheric constituents' concentrations. After the presentation of the full data series for both particles and vapors I would like to see a summarizing Figure for the annual variability of each parameter so that the reader can have a direct impression about the seasonality observed. With regard to diurnal variabilities presented, I recommend normalized rather than absolute values to be used. Additionally, tables with descriptive statistics should

be included.

Specific comments:

Line 169: The red plusses do not contribute to this Figure, they look rather as an apparent red line. A table summarizing what is shown in Figure 2 would be more helpful to understand how much data was eventually excluded from the analysis. What is the benefit from dividing the data in these three categories eventually?

Line 213: Does the comparison with the NAIS refer to the whole measuring period included in this study? If so, what changes for the different instruments used?

Line 325: In Figure 2 the tick marks of the months suggest an earlier start of the season (is it because the mark is set for the mid of the month?).

Line 348: The discussion of Figure 5 is not clear to me. In all data figure, 1.3-1.7nm do not show any variability. 1.1-1.3 nm present a clear maximum in the evening, but only a weak maximum at noon. 1.7-2.5 nm show a clear minimum during night-before sunrise, higher concentrations during the day and maximum values during the evening. I recommend for all diurnal patterns and given the seasonality presented in Figure 6 to present normalized diurnal patterns so that these variabilities become more evident. I also think that there are extremely low values at 00:00 especially for the 1.3-1.7 nm mode that need to be double checked, the rapid drop at NPF days might result from not valid data-it definitely looks weird.

Line 380: No, it does not, there is a single maximum in the evening. Once again I recommend normalizing the values, so that any variability becomes more evident.

Line 385: It looks quite similar to spring to me. A table with statistics of the various concentrations described would be most helpful for the reader to understand the variations described.

Line 407: I see a lot of outliers for high values but none for low values. Once again the pluses are not contributing to the discussion of the Figures, they are rather confusing.

To me, there are pluses that are within the 95th percentile, it has to be clearer.

Line 419: Once again I believe that normalizing the data will produce much better Figures. Additionally, the authors could consider presenting in the same figure the various vapors to assist the eye to identify the variations during the various seasons.

Line 421: Figures 8 and 9 are mixed up, 9 comes before 8, probably in the text as well.

Line 459: It does not look that "sharp".

Line 462: Tables with descriptive statistics will provide the reader a more quantitative perspective of the Figures.

Line 505: This is true for the event periods. However, it is worth mentioning that all vapors have statistically significant correlation with the lower size bins for all data, which implies that these vapors play a significant role in the formation of clusters.

Line 546: What about the correlations described earlier, are they logarithmic as well?

Line 552: HOM Nitrate dimers.

Technical corrections:

Line 39: "in the atmosphere" repetition.

Line 43: Period before "Studies".

Line 144: Period mark repetition.

Line 144: Is it perhaps "thought" rather than "though"?

Line 301: It should be Table 1.

Line 508: Remove paragraph.

Line 558: Remove "correlations".

---

## Referee Comment (RC2) · Anonymous Referee #2 · 7 Sep 2020

The paper by Sulo and co-workers presents long term measurements of sub-3 nm particle concentration and their precursors conducted in Hyytiälä, in the boreal forest. The first part of the study focuses on the identification of optimal settings of the PSM (used for particle measurement) for this site. The second part is dedicated to the study of the time series, including diurnal cycles, of the gas and particle concentrations. The involvement of the selected vapours in the formation of sub-3 nm particles is finally addressed in a last part by the mean of a correlation analysis.

While the data set used is of undeniable value and the objectives presented are of obvious interest, I am however reserved on certain aspects of this study. My main concern is about the definition of the particle size classes used for the analysis, which seem to me too fine in view of the uncertainties associated with the measurement, with a probable impact on the results presented, and in particular on the correlations. Moreover, it is sometimes difficult to extract the main messages from the second part of the study, which is very descriptive, and which I believe would benefit from being sometimes more synthetic. The integration of a more "chemical" dimension to the analysis proposed in Sect. 3.2 would finally, in my opinion, make this second part more complete. These different aspects are presented in more detail in the comments listed below, which I think should be considered before publication of this work.

P2, Introduction: measurements performed in Hyytiälä have enabled numerous studies to be carried out, in particular on the understanding of new particle formation and the identification of its precursors. I would thus suggest to include in the introduction a paragraph recalling some key results specific to this site in order to better situate the objectives and interest of this new study in relation to past work.

P6, L177: The authors indicate the appropriate settings for the station of interest but it is not completely clear to me to which extent these settings are site specific. Could the authors add a sentence or two to briefly comment on these aspects, and discuss in particular the possibility of extrapolating the results obtained to other sites, under what conditions?

P7, Measurement uncertainties: measurement uncertainties related to the nature of the particles and sampling conditions have been the subject of various studies in recent years and are clearly recalled here. Given these uncertainties, I wonder what is the relevance of size classes as fine as those proposed in this work. In particular, the width of the proposed bins is of the same order or less than the uncertainty related to the chemical composition of the particles or their charge. My interrogations are reinforced by the fact that on NPF event days (Fig. 5), the evolution of the concentrations does not seem to show any growth link between the different classes, or at least between the 2 last ones which are considered to be more connected to NPF. I think it would therefore be more appropriate to reduce the number of classes.

P10, L254-256: is the frequency of events of marked stratification known, significant? Should the correlation analysis reported in Sect. 3.3 be limited to day time in the "All data" cases?

P11, L280-282: "Correlations were also separately investigated for spring- and summertime NPF events. There were not enough data points for events during autumn and winter for separate analysis during those seasons." If I am not mistaken, there are correlations reported for autumn in Table 3.

P11, L286: Could the authors add a few words on the value of distinguishing between nitrates and non-nitrates?

P12-13: Time series of the particle concentration:

- ➢ L320-321: "We observe a clear annual maximum during late spring and early summer". I would say that this statement is too strong since it seems to me that it is only verified for 2 years (2016 and 2018). In 2015 and 2019, the concentrations measured in autumn are of the same order as those measured during late spring / early summer, and in 2017, despite the lack of data, it seems that the autumn levels are even higher than those of the previous months.

> L324-332: "Excluding this part of the data did not have a significant impact on the rest of the analysis." Does this mean that the data were effectively excluded for the rest of the analysis?

> L336-337: "and because their data was not filtered to remove scans with too high background". The difference in concentration between the two studies is relatively large (almost an order of magnitude), and I am not sure that the proposed hypotheses can explain such differences. In particular, Fig. 1 suggests that scans with a high background are not systematically associated with higher concentrations than those associated with lower backgrounds (or is it only true for the smallest particles, i.e. in the class 1.1-1.3 nm ?). Was the background itself subtracted from the data in Kontkanen et al. (2017)? Also, I think that based on the studies by Lehtipalo et al. (2014) and Cai et al. (2019), it cannot be excluded that the use of methods other than Kernel (e.g. step wise) could have contributed to the observed differences as well; however, unless I am mistaken, the inversion method used by Kontkanen and co-workers is not specified in their paper.

P14-15: diurnal cycles of the cluster concentration

> The use of a logarithmic scale makes the identification of certain maxima / minima very difficult!

> L369-372: I do not think that the peak observed on event days around noon in the size range 1.7-2.5 nm can be described as a strong maximum. On the other hand, the link between these observations and the occurrence of regional NPF events does not seem obvious to me either; I would expect in this case a chronology in the increase of concentrations (i.e 1.3-1.7 nm and then 1.7-2.5 nm) that is not seen here. Is it also possible that this unexpected chronology is linked to measurement uncertainties / definition of the size classes?

P16: diurnal cycles of the cluster concentration in each season

> The end of this section is very descriptive, with observations that are often difficult to confirm due to the logarithmic scale (e.g. L389), and it is globally difficult for me to extract a message from this analysis. I would suggest to add one or two sentences at the end of the section to summarize the main outcomes.

> Concentration levels and the presence or absence of distinctive diurnal cycles are primarily related to the frequency of occurrence of NPF in each season. However, this explanation does not seem to be sufficient, since although the frequencies of NPF are lowest in winter, the concentrations observed are on average higher than those in autumn for the 2 size classes 1.3-1.7 and 1.7-2.5 nm, and comparable or even higher than those in summer for the class 1.3-1.7 nm.

P17-20: Time series of vapour concentrations:

> This section is once again very descriptive, with an analysis that sometimes seems too superficial to me. It would, in my opinion, benefit from being more detailed from a chemical point of view, particularly with regard to the processes involved in the formation/transformation of the organic compounds of interest according to their nature (monomers vs. dimers, nitrates vs. non-nitrates) and their influence on the cycles observed. The findings of studies previously dedicated to the specific analysis of cluster composition at the site (e.g. Yan et al. 2016; Bianchi et al., 2017; Rose et al. 2018 and connected literature) could for instance be explicitly mentioned to benefit the interpretation of the results.

➤ L406: "during summer months": based on Fig. 7, it should in my opinion be changed to "between late spring and early autumn".

➤ L445-450: The importance of SA and HOM non-nitrate monomers in the formation and initial growth of the clusters cannot be deduced with such level of confidence from the comparison of median diurnal cycles alone ("This points to the importance", "implying that"). Based on Fig. 5, the other selected compounds also have variations that are comparable to that of the cluster concentration, and could thus be involved in their formation as well (HOM nitrate monomers peaking around midday, dimers peaking in the evening). I would suggest using more moderate phrasing at this stage of the analysis.

➤ L451-487: As in the previous section, I find it difficult to extract the main messages from this relatively long description. I would suggest, for example, to first highlight the fact that the general pattern of the cycles is overall comparable for all seasons, but with amplitudes and/or periods that are variable and probably modulated by the amount of global radiation; I would then describe the main differences observed between each season.

P21-25: connection between precursor vapours and cluster concentrations

The search for correlations between the presence of newly formed particles and possible precursors seems to me relevant for the identification of the compounds involved in the formation and early growth process. However, I believe that the uncertainties related to the definition of such fine particle size classes as those used here limit the relevance of such correlation study, whose results are for me difficult to interpret. I still have a few comments on this section:

➤ L507: I would suggest to better explain the choice of the combinations of compounds to be investigated.

➤ L515: "However, the difference in correlation coefficients is not large": It depends on the meaning of large but I would say that the difference is sometimes significant!

➤ L520-521: "However, the correlation can also point to two separate formation pathways, organic and inorganic." I would suggest to mention the earlier results of Yan et al. (2018) to support the idea of multiple mechanisms.

➤ Fig. 10: Line 494-495 indicates that calibration coefficients are not considered in the correlation analysis, but they seem to be accounted for in Fig. 10. Furthermore, the orders of magnitude shown on the x-axis in the third column (product of the previous 2 columns) do not seem to be consistent with what is shown in the previous columns.

P25-26 : Conclusions

Beyond the fact that it remains difficult to approve certain observations due to the use of logarithmic scales, some of the information that is recalled here does not seem to me to be precise enough, or not in line with what is reported in the previous sections:

- L592: "The 1.3–1.7 nm and 1.7–2.5 nm particle concentrations show a marked increase during springtime": this is not obvious for 1.7-2.5 nm particles, whose concentration is close to what is observed in summer.

- L593-594: "The diurnal patterns of sub-3nm concentrations exhibit clear daytime maxima around midday.": It does not seem to be true for all sizes, seasons, types of days (NPF vs non event days).
- L594: "This maximum is the clearest during spring and autumn": Barring any misunderstanding on my part, this message seems to me to contradict what is reported in L393-396 regarding autumn: "weak diurnal pattern", "no discernible diurnal pattern".

Technical / Minor comments

P2, L43: point missing at the end of the sentence.

P3, L79: "for measuring particle concentrations larger than 1 nm in size": I would suggest to change to "for measuring the concentration of particles larger than 1 nm in size".

P5, L144: extra point at the end of the sentence.

P11, L278: extra "then".

P13, L334: "concentration" should be removed at the end of the sentence.

P19, L460: point missing at the end of the sentence.

P24, L558: "correlations" should be removed.

Figures:

➤ I would suggest homogenising, as far as possible, the appearance of the figures and in particular the font size used. Adding a grid would also make the values easier to read.

➤ Fig. 5: the scale of the ordinate axis (left) of the third panel should be slightly adjusted to match the scale of the others and ease the comparison.

The writing of times should be harmonized (e.g. L389 vs L424).

---

## Author Comment (AC1) · 2 Nov 2020

In this manuscript, five years of sub-3 nm particle concentrations obtained with the PSM and three years of precursor vapor concentrations measured with the CI-APi-ToF at the boreal environment are presented, and possible correlation between particles and vapors are examined. This is a unique dataset analyzed and the results are definitely worth being presented in ACP after few minor revisions.

A general comment is the presentation of variabilities of the various atmospheric constituents' concentrations. After the presentation of the full data series for both particles and vapors I would like to see a summarizing Figure for the annual variability of each parameter so that the reader can have a direct impression about the seasonality observed. With regard to diurnal variabilities presented, I recommend normalized rather than absolute values to be used. Additionally, tables with descriptive statistics should C1 be included.

As per the reviewer's suggestion, we have added annual variability figures for both data series and added a summary table of descriptive statistics. We have also changed the diurnal variability figures so that each particle size range and vapor has its own figure in linear scale, allowing the reader to observe the diurnal patterns clearly while keeping the absolute values. We feel that comparing the absolute values from the figures at a glance is an important part of the figure. This also answers the comment by reviewer 2.

Specific comments:

Line 169: The red plusses do not contribute to this Figure, they look rather as an apparent red line. A table summarizing what is shown in Figure 2 would be more helpful to understand how much data was eventually excluded from the analysis. What is the benefit from dividing the data in these three categories eventually?

We have excluded the outliers (red plusses) from Figure 2 and added a table to summarize the amount of data in each category. The reasons for dividing the data into these categories are detailed in L170-180 "When the background is under 1 cm-3, the measured concentrations are on average lower than when the background level is above 1 cm-3, indicating that we are not activating all of the 1.1 - 1.7 nm particles at those settings. However, if the background level rises to over 10 cm-3, also the variation becomes notably larger, underlining the various factors affecting the concentration at higher background. At high background levels the PSM likely activates large vapor molecules or clusters whose concentrations are not stable, leading to larger variation in the concentration. When these species dominate the activated particles, the particle size distribution cannot be easily resolved from the scans.."

Line 213: Does the comparison with the NAIS refer to the whole measuring period included in this study? If so, what changes for the different instruments used?

No, it does not, as the neutral PSM needed for this analysis was only available from 2017 to 2019. Added the line "The ion concentrations were acquired from a PSM with an ion trap inlet (Wagner et al., 2017, Kangasluoma et al., 2016a) measuring in the same container as the long-term measurement PSM. The neutral PSM data was measured between April 2017 and April 2019."

Line 325: In Figure 2 the tick marks of the months suggest an earlier start of the season (is it because the mark is set for the mid of the month?).

We assume the reviewer means Figure 4 in this comment. The ticks for the boxplots are set in the middle of the month. Clarified the caption by adding the line "The tick marks visible are in the middle of the starting month for each season."

Line 348: The discussion of Figure 5 is not clear to me. In all data figure, 1.3-1.7nm do not show any variability. 1.1-1.3 nm present a clear maximum in the evening, but only a weak maximum at noon. 1.7-2.5 nm show a clear minimum during night-before sunrise, higher concentrations during the day and maximum values during the evening. I recommend for all diurnal patterns and given the seasonality presented in Figure 6 to present normalized diurnal patterns so that these variabilities become more evident. I also think that there are extremely low values at 00:00 especially for the 1.3-1.7 nm mode that need to be double checked, the rapid drop at NPF days might result from not valid data-it definitely looks weird.

We changed the diurnal pattern figures to include one size range per figure in order to better observe the diurnal variation and maximums and minimums. We also changed the data to only two size bins as suggested by reviewer 2 to reduce the uncertainty from the sizing in the inversion. The data points were also double checked. Following these changes, we revised the text discussing Figure 6 (old Fig 5.).

Line 380: No, it does not, there is a single maximum in the evening. Once again I recommend normalizing the values, so that any variability becomes more evident.

This has been amended in the text: "We observe a maximum for the 1.1–1.7 nm concentration in the evening and a second smaller peak in the afternoon."

Line 385: It looks quite similar to spring to me. A table with statistics of the various concentrations described would be most helpful for the reader to understand the variations described.

We have added a table with descriptive statistics of both particle and vapor concentrations.

Line 407: I see a lot of outliers for high values but none for low values. Once again the pluses are not contributing to the discussion of the Figures, they are rather confusing. C2 To me, there are pluses that are within the 95th percentile, it has to be clearer.

We have clarified the figure by removing the outlier values from the figure to clarify the discussion.

Line 419: Once again I believe that normalizing the data will produce much better Figures. Additionally, the authors could consider presenting in the same figure the various vapors to assist the eye to identify the variations during the various seasons.

We have elected to represent each vapor in its own figure to better allow a comparison between different seasons and between NPF and non-event days and changed the scale from logarithmic to linear. We believe the figures are much clearer now.

Line 421: Figures 8 and 9 are mixed up, 9 comes before 8, probably in the text as well.

This has been corrected and the Figure 8 now comes before Figure 9. The numbering of the figures has changed due to the annual variation figures added to the manusctipt.

Line 459: It does not look that "sharp".

We have rewritten this section to be more synthetic and it does not describe the peak as sharp anymore.

Line 462: Tables with descriptive statistics will provide the reader a more quantitative perspective of the Figures.

A table of descriptive statistics depicting the medians and the $25^{th}$ and $75^{th}$ percentiles has been added to the manuscript (Table 3).

Line 505: This is true for the event periods. However, it is worth mentioning that all vapors have statistically significant correlation with the lower size bins for all data, which implies that these vapors play a significant role in the formation of clusters.

The new results from splitting the data into two size bins have changed these results, and although the results still show a statistically significant correlation, the correlation coefficients themselves are lower. This is likely due to other dynamics included in the lowest size bin due to its expansion. We added a sentence saying: "Particle concentrations show positive correlation with all the measured precursor vapors, but we can see slight differences between the two size ranges and between NPF event times and the whole data set."

Line 546: What about the correlations described earlier, are they logarithmic as well?

Yes, they are. This has been checked and the term logarithmic added to appropiate sections when the correlations are mentioned.

Line 552: HOM Nitrate dimers.

Added the word "nitrate".

Technical corrections:

Line 39: "in the atmosphere" repetition.

Removed the repetition.

Line 43: Period before "Studies".

Added a period.

Line 144: Period mark repetition.

Removed the repetition.

Line 144: Is it perhaps "thought" rather than "though"?

The reviewer is correct. This has been amended.

Line 301: It should be Table 1.

Corrected to "Table 1".

Line 508: Remove paragraph.

Paragraph removed.

Line 558: Remove "correlations".

Removed "correlations".

Reviewer 2:

The paper by Sulo and co-workers presents long term measurements of sub-3 nm particle concentration and their precursors conducted in Hyytiälä, in the boreal forest. The first part of the study focuses on the identification of optimal settings of the PSM (used for particle measurement) for this site. The second part is dedicated to the study of the time series, including diurnal cycles, of the gas and particle concentrations. The involvement of the selected vapours in the formation of sub-3 nm particles is finally addressed in a last part by the mean of a correlation analysis.

While the data set used is of undeniable value and the objectives presented are of obvious interest, I am however reserved on certain aspects of this study. My main concern is about the definition of the particle size classes used for the analysis, which seem to me too fine in view of the uncertainties associated with the measurement, with a probable impact on the results presented, and in particular on the correlations. Moreover, it is sometimes difficult to extract the main messages from the second part of the study, which is very descriptive, and which I believe would benefit from being sometimes more synthetic. The integration of a more "chemical" dimension to the analysis proposed in Sect. 3.2 would finally, in my opinion, make this second part more complete. These different aspects are presented in more detail in the comments listed below, which I think should be considered before publication of this work.

P2, Introduction: measurements performed in Hyytiälä have enabled numerous studies to be carried out, in particular on the understanding of new particle formation and the identification of its precursors. I would thus suggest to include in the introduction a paragraph recalling some key results specific to this site in order to better situate the objectives and interest of this new study in relation to past work.

Good suggestion. Added a short paragraph recalling some of the key results from earlier studied from the SMEAR II data into the introduction: "The long-term measurements at the SMEAR II station in Hyytiälä, Finland, have enabled studying atmospheric new particle formation and its prerequisites. Prior research has investigated the frequency of NPF in the boreal forest (Dal Maso et al. 2005; Nieminen et al., 2014) and how it is affected by condensation sink and cloudiness (Dada et al., 2017) and other meteorological conditions (Sogacheva et al., 2007). These studies have

concluded that NPF is most common in Hyytiälä during spring and that NPF occurs most often during days with fewer clouds and a low condensation sink. Shorter campaign measurements have been used to investigate the connection between NPF and its precursors vapors (e.g. Riipinen et al. 2007; Kulmala et al., 2013; Yan et al. 2018). They indicate that both sulfuric acid and some organic vapors participate in NPF in Hyytiälä, but the possible seasonality of the mechanism and exact identity of the organic compounds are still unclear.

P6, L177: The authors indicate the appropriate settings for the station of interest but it is not completely clear to me to which extent these settings are site specific. Could the authors add a sentence or two to briefly comment on these aspects, and discuss in particular the possibility of extrapolating the results obtained to other sites, under what conditions?

We have added a comment on the suitability of these settings for other measurement locations: "The optimal settings determined here can likely be used in measurement sites with similar particle concentrations and composition. However, if a measurement location has much higher particle concentrations or the changes in particle concentration are much more rapid than in a boreal forest, for examples in an urban measurement location, they will have to be adjusted. Also, the composition of smallest particles might affect the ideal level of supersaturation to activate most of them without too much disturbance from homogenous nucleation."

P7, Measurement uncertainties: measurement uncertainties related to the nature of the particles and sampling conditions have been the subject of various studies in recent years and are clearly recalled here. Given these uncertainties, I wonder what is the relevance of size classes as fine as those proposed in this work. In particular, the width of the proposed bins is of the same order or less than the uncertainty related to the chemical composition of the particles or their charge. My interrogations are reinforced by the fact that on NPF event days (Fig. 5), the evolution of the concentrations does not seem to show any growth link between the different classes, or at least between the 2 last ones which are considered to be more connected to NPF. I think it would therefore be more appropriate to reduce the number of classes.

As per the reviewer's suggestion, we have reduced the number of size classes to two – 1.1-1.7 nm and 1.7-2.5 nm. One for the smallest particles with more constant behaviour and one for particles that are clearly connected to NPF. With this new division the growth link becomes very clear in Fig 6: the 1.1-1.7 nm particles start rising on NPF event days a little bit earlier than the 1.7-2.5 nm particles.

P10, L254-256: is the frequency of events of marked stratification known, significant? Should the correlation analysis reported in Sect. 3.3 be limited to day time in the "All data" cases?

According to Alekseychick et al. (2013) ca. 19% of nights show strong stratification (decoupling of the sub-canopy and above-canopy layers) in Hyytiälä. Since it is a minority of data, and because we believe it is important to include the nighttime to get more variability in the concentrations, we decided to include also nighttime into the correlation analysis. Furthermore, we investigated the correlations with the "All data" cases limited to daytime and the difference in correlations was minor (+- 0.05) for all investigated vapors and particle size ranges.

P11, L280-282: "Correlations were also separately investigated for spring- and summertime NPF events. There were not enough data points for events during autumn and winter for separate analysis during those seasons." If I am not mistaken, there are correlations reported for autumn in Table 3.

The reviewer is correct and this has been amended in the text: "Correlations were also separately investigated for spring- and summer- and autumntime NPF events."

P11, L286: Could the authors add a few words on the value of distinguishing between nitrates and nonnitrates?

We have added a sentence explaining the value of distinguishing between nitrates and non-nitrates: "The molecules were divided into nitrates and non-nitrates because nitrate HOMs typically have a higher volatility than non-nitrate molecules (Yan et al., 2020) and it is possible that their contribution to NPF is different (Lehtipalo et al. 2018)."

P12-13: Time series of the particle concentration:
     ☐ L320-321: "We observe a clear annual maximum during late spring and early summer". I would say that this statement is too strong since it seems to me that it is only verified for 2 years (2016 and 2018). In 2015 and 2019, the concentrations measured in autumn are of the same order as those measured during late spring / early summer, and in 2017, despite the lack of data, it seems that the autumn levels are even higher than those of the previous months.

We have removed the word "clear" from the statement and added a figure to better illustrate the median annual variability.

     ☐ L324-332: "Excluding this part of the data did not have a significant impact on the rest of the analysis." Does this mean that the data were effectively excluded for the rest of the analysis?

The data was not excluded, this was just a test. For clarity, this sentence was deleted entirely, since we did not end up excluding this part of the data.

     ☐ L336-337: "and because their data was not filtered to remove scans with too high background". The difference in concentration between the two studies is relatively large (almost an order of magnitude), and I am not sure that the proposed hypotheses can explain such differences. In particular, Fig. 1 suggests that scans with a high background are not systematically associated with higher concentrations than those associated with lower backgrounds (or is it only true for the smallest particles, i.e. in the class 1.1-1.3 nm ?). Was the background itself subtracted from the data in Kontkanen et al. (2017)? Also, I think that based on the studies by Lehtipalo et al. (2014) and Cai et al. (2019), it cannot be excluded that the use of methods other than Kernel (e.g. step wise) could have contributed to the observed differences as well; however, unless I am mistaken, the inversion method used by Kontkanen and co-workers is not specified in their paper.

Figure 1 shows the fraction of bad scans and does not appear to suggest what the reviewer is saying. Assuming that the reviewer is referring to Figure 2, we still see a higher portion of high concentration values in scans with high background, even if they are not systematically associated with those high background scans. we have specified a further reason in the text – in that Kontkanen et al. (2017) subtracted the background from the data manually – and the background measurements were conducted manually (and thus less frequently), resulting in a possible underestimation of the background. .However, the exact reason for the difference is not clear and might be a combination of true variability between years and technical reasons. Considering that the uncertainties in sub-3nm measurements (especially in the detection efficiency of neutral particles) are still significant, the most important aspect in this study are not the absolute concentrations but the diurnal and annual variation in different size classes and correlations with the precursors.

P14-15: diurnal cycles of the cluster concentration

The use of a logarithmic scale makes the identification of certain maxima / minima very difficult!

This has been amended and the diurnal cycle figures are now in linear scale.

 L369-372: I do not think that the peak observed on event days around noon in the size range 1.7-2.5 nm can be described as a strong maximum. On the other hand, the link between these observations and the occurrence of regional NPF events does not seem obvious to me either; I would expect in this case a chronology in the increase of concentrations (i.e 1.3-1.7 nm and then 1.7-2.5 nm) that is not seen here. Is it also possible that this unexpected chronology is linked to measurement uncertainties / definition of the size classes?

The reviewer is correct that the unxpected chronology could be due to the definition of the size classes. We have reduced our size classes to two in order to reduce the uncertainties in the sizing.

P16: diurnal cycles of the cluster concentration in each season  The end of this section is very descriptive, with observations that are often difficult to confirm due to the logarithmic scale (e.g. L389), and it is globally difficult for me to extract a message from this analysis. I would suggest to add one or two sentences at the end of the section to summarize the main outcomes.
Concentration levels and the presence or absence of distinctive diurnal cycles are primarily related to the frequency of occurrence of NPF in each season. However, this explanation does not seem to be sufficient, since although the frequencies of NPF are lowest in winter, the concentrations observed are on average higher than those in autumn for the 2 size classes 1.3- 1.7 and 1.7-2.5 nm, and comparable or even higher than those in summer for the class 1.3-1.7 nm.

The seasonal diurnal cycle section has been rewritten to be more synthetic. The differences in concentration are more visible, and the winter concentrations are the lowest. We have also added text to explain the evening maximums.

P17-20: Time series of vapour concentrations:

 This section is once again very descriptive, with an analysis that sometimes seems too superficial to me. It would, in my opinion, benefit from being more detailed from a chemical point of view, particularly with regard to the processes involved in the formation/transformation of the organic compounds of interest according to their nature (monomers vs. dimers, nitrates vs. non-nitrates) and their influence on the cycles observed. The findings of studies previously dedicated to the specific analysis of cluster composition at the site (e.g. Yan et al. 2016; Bianchi et al., 2017; Rose et al. 2018 and connected literature) could for instance be explicitly mentioned to benefit the interpretation of the results.

We have added more interpretation from a chemical point of view, speculating about the sources of HOM monomers and dimers as well nitrates and non-nitrates.

 L406: "during summer months": based on Fig. 7, it should in my opinion be changed to "between late spring and early autumn".

This has been amended as the reviewer as suggested.

 L445-450: The importance of SA and HOM non-nitrate monomers in the formation and initial growth of the clusters cannot be deduced with such level of confidence from the comparison of median diurnal cycles alone ("This points to the importance", "implying that"). Based on Fig. 5,

the other selected compounds also have variations that are comparable to that of the cluster concentration, and could thus be involved in their formation as well (HOM nitrate monomers peaking around midday, dimers peaking in the evening). I would suggest using more moderate phrasing at this stage of the analysis.

We have amended the language in this section to be more moderate: "During regional NPF event days, the concentrations of all analyzed aerosol precursor vapors are higher than during non-event days (Figure 10). However, the diurnal patterns of the precursor vapors are otherwise rather similar on event and non-event days. The non-nitrate monomers and dimers show a small increase during daytime on NPF days which is absent on diurnal patterns of global radiation in Figures 6 and 9, suggest that during event days there is more photochemical production and potentially also higher emissions of biogenic vapors. However, from the diurnal variations is not possible point out a single vapor explaining the difference between NPF events and non-events."

☐ L451-487: As in the previous section, I find it difficult to extract the main messages from this relatively long description. I would suggest, for example, to first highlight the fact that the general pattern of the cycles is overall comparable for all seasons, but with amplitudes and/or periods that are variable and probably modulated by the amount of global radiation; I would then describe the main differences observed between each season.

We have rewritten this section to be more synthetic and highlighted the effect of global radiation.

P21-25: connection between precursor vapours and cluster concentrations The search for correlations between the presence of newly formed particles and possible precursors seems to me relevant for the identification of the compounds involved in the formation and early growth process. However, I believe that the uncertainties related to the definition of such fine particle size classes as those used here limit the relevance of such correlation study, whose results are for me difficult to interpret. I still have a few comments on this section:

☐ L507: I would suggest to better explain the choice of the combinations of compounds to be investigated.

This has been explained in more detail in Chapter 2.5 now with some added detail: ". Laboratory experiments replicating boundary layer NPF in forested regions (Riccobono et al., 2014; Lehtipalo et al., 2018) and previous analysis of field data sets (Paasonen et al., 2010) have shown that particle formation rates can be parametrized using a product of sulfuric acid concentration and organics concentrations. Therefore, we expect that sulfuric acid and HOMs or some combination of them should correlate well with new particle formation in a boreal forest and the correlations in different size ranges might reveal which precursors are most important at different stages of the process."

☐ L515: "However, the difference in correlation coefficients is not large": It depends on the meaning of large but I would say that the difference is sometimes significant!

We have made the comment more specific with "However, the difference in coefficients is not very large with the uncertainties involved"

☐ L520-521: "However, the correlation can also point to two separate formation pathways, organic and inorganic." I would suggest to mention the earlier results of Yan et al. (2018) to support the idea of multiple mechanisms.

Added a reference to Yan et al. (2018).

☐ Fig. 10: Line 494-495 indicates that calibration coefficients are not considered in the correlation analysis, but they seem to be accounted for in Fig. 10. Furthermore, the orders of magnitude shown on the x-axis in the third column (product of the previous 2 columns) do not seem to be consistent with what is shown in the previous columns.

This was indeed an error in the plot, but this has now been rectified and the new Figure shows the data without the calibration coefficients and with the correct magnitudes.

P25-26 : Conclusions Beyond the fact that it remains difficult to approve certain observations due to the use of logarithmic scales, some of the information that is recalled here does not seem to me to be precise enough, or not in line with what is reported in the previous sections:
- L592: "The 1.3–1.7 nm and 1.7–2.5 nm particle concentrations show a marked increase during springtime": this is not obvious for 1.7-2.5 nm particles, whose concentration is close to what is observed in summer.

We have softened the language to The 1.7–2.5 nm particle concentration, on the other hand, exhibits the largest median concentration during springtime, coinciding with increased regional NPF frequency." and this is reflected in the Results section as well. The median concentrations are in Table 2.

- L593-594: "The diurnal patterns of sub-3nm concentrations exhibit clear daytime maxima around midday.": It does not seem to be true for all sizes, seasons, types of days (NPF vs non event days).

This has been amended to "The daytime maximum in concentrations is clear during spring, summer and autumn, during which regional NPF events are also more common."

- L594: "This maximum is the clearest during spring and autumn": Barring any misunderstanding on my part, this message seems to me to contradict what is reported in L393-396 regarding autumn: "weak diurnal pattern", "no discernible diurnal pattern".

The new figures present the diurnal pattern more clearly and this is reflected both in the conclusions and in the results.

Technical / Minor comments

P2, L43: point missing at the end of the sentence.

Corrected.

P3, L79: "for measuring particle concentrations larger than 1 nm in size": I would suggest to change to "for measuring the concentration of particles larger than 1 nm in size".

We have changed the sentence to the suggested "for measuring the concentration of particles larger than 1 nm in size".

P5, L144: extra point at the end of the sentence.

Corrected.

P11, L278: extra "then".

Corrected.

P13, L334: "concentration" should be removed at the end of the sentence.

Corrected.

P19, L460: point missing at the end of the sentence.

This paragraph has been entirely rewritten.
P24, L558: "correlations" should be removed.

Removed.

Figures: □
    I would suggest homogenising, as far as possible, the appearance of the figures and in particular the font size used. Adding a grid would also make the values easier to read.

    □ Fig. 5: the scale of the ordinate axis (left) of the third panel should be slightly adjusted to match the scale of the others and ease the comparison.

The new figure is easier to read and the concentrations are easier to read.

The writing of times should be harmonized (e.g. L389 vs L424).

Times have all been corrected to the format HH:MM.

References:

Alekseychik, P., Mammarella, I., Launiainen, S., Rannik, Ü., and Vesala, T.: Evolution of the nocturnal decoupled layer in a pine forest canopy, Agr. Forest. Meteorol., 174, 15–27, https://doi.org/10.1016/j.agrformet.2013.01.011, 2013.